



# An enhanced integrated water vapour dataset from more than 10,000 global ground-based GPS stations in 2020

Peng Yuan[1], Geoffrey Blewitt[2], Corné Kreemer[2], William C. Hammond[2], Donald Argus[3], Xungang Yin[4], Roeland Van Malderen[5], Michael Mayer[1], Weiping Jiang[6], Joseph Awange[7], Hansjörg Kutterer[1]

5   1 Karlsruhe Institute of Technology, Geodetic Institute, Karlsruhe, BW, Germany

2 Nevada Bureau of Mines and Geology, University of Nevada, Reno, NV, USA

3 Jet Propulsion Laboratory, California Institute of Technology, Pasadena, CA, USA

4 NOAA National Centers for Environmental Information, Asheville, NC, USA

5 Royal Meteorological Institute of Belgium, Brussels, Belgium

10  6 Wuhan University, GNSS Research Center, Wuhan, China

Curtin University, School of Earth and Planetary Sciences, Perth, Australia

*Correspondence to*: Peng Yuan (peng.yuan@kit.edu)

**Abstract.** We developed a high-quality global Integrated Water Vapour (IWV) dataset from 12,552 ground-based Global Positioning System (GPS) stations in 2020. It consists of 5-min GPS IWV estimates with a total number of 1,093,591,492

datapoints. The completeness rates of the IWV estimates are higher than 95% at 7253 (58%) stations. The dataset is an enhanced version of the existing operational GPS IWV dataset provided by Nevada Geodetic Laboratory (NGL). The enhancement is reached by employing accurate meteorological information from the fifth generation of European ReAnalysis (ERA5) for the GPS IWV retrieval with a significantly higher spatiotemporal resolution. A dedicated data screening algorithm is also implemented. The GPS IWV dataset has a good agreement with in-situ radiosonde observations

at 182 collocated stations worldwide. The IWV biases are within ±3.0 kg m$^{-2}$ with a Mean Absolute Bias (MAB) value of 0.69 kg m$^{-2}$. The standard deviations (SD) of IWV differences are no larger than 3.4 kg m$^{-2}$. In addition, the enhanced IWV product shows substantial improvements compared to NGL's operational version, and it is thus recommended for high-accuracy applications, such as research of extreme weather events and diurnal variations of IWV, and intercomparisons with other IWV retrieval techniques. Taking the radiosonde-derived IWV as reference, the MAB and SD of IWV differences are

reduced by 19.5% and 6.2% on average, respectively. The number of unrealistic negative GPS IWV estimates are also substantially reduced by 92.4% owing to the accurate Zenith Hydrostatic Delay (ZHD) derived by ERA5. The dataset is available at: https://doi.org/10.5281/zenodo.6973528 (Yuan et al., 2022).



# 1 Introduction

Water vapour plays a significant role in various Earth system processes. As the medium of moisture and latent heat in the atmosphere, water vapour affects many meteorological, climatological, and hydrological phenomena in a range of spatial and temporal scales, such as extreme weather (e.g., Zhu and Newell, 1994), climate change (e.g., Karl and Trenberth, 2003; Yuan et al., 2021a), and water cycle (e.g., Worden et al., 2007). Global atmospheric water vapour has been very likely increasing due to global warming since the 1980s as reported by IPCC 6th Assessment Report (Masson-Delmotte et al.,
2021). It thus tends to aggravate various extreme weather and climate events, such as floods (e.g., Turato et al., 2004) and droughts (e.g., Jiang et al., 2017; Awange 2022). Therefore, analysis of atmospheric water vapour is essential for deepening our understanding of the Earth system.

 Accurate water vapour measurement with a high spatiotemporal resolution remains a challenge due to its large variability in both space and time (Jin et al., 2008; Vogelmann et al., 2015). The atmospheric water vapour can be quantified
as Integrated Water Vapour (IWV), which is the integrated mass of water vapour in a vertical atmospheric column over a unit area (in unit of kg m$^{-2}$). Several global IWV datasets have been developed with various techniques recently, such as satellite (Beirle et al., 2018) and radiosonde (Ferreira et al., 2019) measurements. However, these instruments have their respective strengths and shortcomings. For instance, satellite measurements have good spatial coverage, but their spatiotemporal resolutions could be low. Radiosondes can provide precise vertical distribution of water vapour, but their
spatial densities and temporal resolutions are low.

 Ground-based Global Positioning System (GPS) has become a unique IWV retrieval technique with outstanding features like high accuracy, high temporal resolution, and all-weather capability (Jones et al., 2020). As the most prevalent Global Navigation Satellite System (GNSS), GPS has been introduced for water vapour monitoring since the early 1990s (Bevis et al., 1992). GPS signals emitted from satellites to ground-based receivers are delayed when passing the Earth's
atmosphere (Awange 2012, 2018). The slant satellite-to-receiver signal delays due to the nondispersive troposphere are mapped into zenith direction and estimated in GPS data processing, termed as Zenith Total Delay (ZTD). The ZTD consists of hydrostatic and wet parts. The Zenith Hydrostatic Delay (ZHD) can be estimated by using pressure observation or obtained from an a priori ZHD product. The Zenith Wet Delay (ZWD) can then be converted into IWV exploiting auxiliary meteorological information. GPS IWV retrievals are not only useful in the analyses of related phenomena but also serve as a
valuable data source for the assessments of other IWV products derived by various techniques (e.g., Van Malderen et al., 2014; Vaquero-Martínez et al., 2018; Yu et al., 2021; Yuan et al., 2021a; Yuan et al., 2021b).

 However, the insufficient spatial densities and coverages of current available GPS IWV datasets severely limit their capabilities in sensing Earth system processes over various spatial scales. Although approximately ten thousand of continuous dual-frequency GPS stations are currently in operation around the world, most existing regional and global GPS
IWV datasets only include several hundred continuous stations or campaign measurements spanning less than one year, such as IGS (Wang et al., 2007; Heise 2009), EUREF (Pacione et al., 2017), HyMeX (Bock et al., 2016), EUREC4A (Bock et al.,



2021), and GURN (Yuan and Kutterer, 2021; Wagner et al., 2022; Fersch et al., 2022). Combining their solutions for an integrated analysis is possible but it could be impacted by systematic differences due to various software packages and strategies used in their GPS data processing as well as approaches adopted for the conversion of IWV. Nevada Geodetic

Laboratory (NGL) has taken a big step forward by integrating global GPS observations at more than 10,000 stations and providing their tropospheric ZTD and IWV estimates as well as other products (Blewitt et al., 2018). Although NGL processes the GPS observations uniformly by employing many state-of-the-art models and strategies, the accuracy of its operational IWV product is limited by the inaccurate auxiliary meteorological information used in IWV retrieval. An enhancement of NGL's operational GPS IWV product will provide more accurate IWV estimates and thus more value for

many applications, such as analysis of extreme weather and diurnal variations of IWV, and assessment of other IWV retrieval techniques.

In this work, we aim to enhance NGL's operational GPS IWV dataset by using high-quality auxiliary meteorological data from the newly released ERA5 atmospheric reanalysis (Hersbach et al., 2020) provided by European Centre for Medium-Range Weather Forecasts (ECMWF) for the conversion of IWV. Section 2 describes the production details of the

enhanced GPS IWV dataset. Section 3 analyses the characteristics of the global IWV time series. Sections 4 and 5 compare the enhanced GPS IWV dataset with respect to radiosonde data and NGL's operational GPS IWV dataset, respectively. Section 6 summarises this work.

## 2 Data and methods

### 2.1 GPS data processing

Nevada Geodetic Laboratory (NGL) collects and processes geodetic-quality GPS observations at more than 10,000 stations worldwide from many regional and commercial networks in addition to the commonly used International GNSS Service (IGS) network. NGL routinely processes the observations by using GipsyX version 1.0 software (Bertiger et al., 2020) released by Jet Propulsion Laboratory (JPL) with precise point positioning (Zumberge et al., 1997). JPL's Repro3 final GPS orbits and clocks are also used. The tropospheric delay is modelled with Vienna Mapping Function 1 (VMF1) and

associated gridded map products of a priori ZHD and ZWD. VMF1 has separate mapping functions for ZHD and ZWD. The products are provided by Vienna University of Technology (TUW, Boehm et al., 2006) based on ECMWF operational analysis. In addition, the asymmetry of the tropospheric delay was modelled with horizontal gradient parameters in north-south and east-west. Biases in the ZWD and two horizontal gradient parameters are estimated as random walk processes with 5-min resolution with constraints of 3 and 0.3 mm h$^{-1/2}$, respectively. This estimation assumes that ZHD is correct, and all the

residual delay is in ZWD. The final ZTD = ZHD + ZWD is not sensitive to small errors in a priori ZHD. The cut-off elevation angle is set to 7° and an elevation ($e$) dependent weighting function of $\sin e$ is adopted. The 1st order ionospheric effect is removed by linear combination of dual-frequency observables. The 2nd order effect is corrected using JPL's IONEX gridded map product and the International Geomagnetic Reference Field 12 (IGRF12; Thébault et al., 2015), which has been

reported to be able to improve the tropospheric estimates (Zus et al., 2017). Absolute IGS antenna phase centre model is

applied (Schmid et al., 2007). Solid Earth tides, oceanic and atmospheric tidal loadings are modelled as recommended by the

International Earth Rotation and Reference Systems Service (IERS) 2010 conventions (Petit and Luzum, 2010), while the

non-tidal oceanic and atmospheric tidal loadings are kept as uncorrected. Simultaneously estimated with the tropospheric

parameters are three-dimensional station coordinates every 24 hours, and a station clock bias every 5 minutes. Integer

ambiguity resolution is applied to estimated carrier phase biases. More details on the data processing strategy are given in

http://geodesy.unr.edu/gps/ngl.acn.txt (last accessed on 2022-06-06).

**Figure 1.** (a) Geographical distribution of the 12,555 GPS stations (cyan dots) and 182 radiosonde stations. (b–d) zoomed in figures for USA, Europe, and Japan, respectively. The radiosonde types are labelled in various colours.





We collected the global ZTD estimates at 12,612 GPS stations in 2020 provided by NGL. We checked the consistency between the actual position of each station and its long-term reference position provided in the station list file. The check is necessary because their reference positions were used to calculate the auxiliary meteorological information, and incorrect IWV estimates could be obtained if the positions are inconsistent. We blacklisted 57 stations as their distances were larger than 10 m in vertical or 100 m in horizontal. The position inconsistencies could be due to station migration (e.g., some of

them are on flowing ice), local relocation of station antennas, or misidentification of duplicated station names.

Figure 1 shows the locations of the utilised 12,555 stations, which are most densely distributed in the USA, Europe, and Japan. We calculated completeness rate for each station, which is the ratio between the number of a given station's ZTD estimates and the full number of 5-min epochs in 2020. Results show that the average completeness rate of the 12,555 stations was 83.1%, and the rates were higher than 95% at 7,414 stations (Fig. 2a). The total number of the 5-min GPS ZTD

estimates are 1,100,442,802, which is equal to 87,650 datapoints per station and 10,440 datapoints every 5 minutes (Fig. 2b). It is noticeable from Fig. 2a that the number of datapoints reduces over time. This is because the data of several regional networks were missing in the second half of the year, such as the South Korean stations.

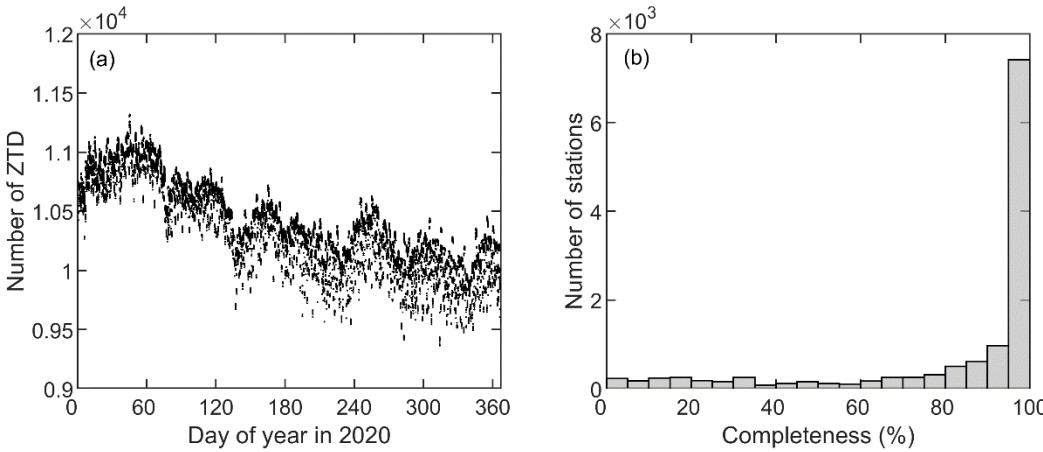

**Figure 2.** (a) The number of raw GPS ZTD estimates of the 12,555 stations at each 5-min epoch. (b) Statistics of
completeness rates of the stations. The completeness rate of a given GPS station was calculated as the ratio between the number of its ZTD estimates and the full number of 5-min epochs in 2020 (i.e., 105,408).

### 2.2 Screening of Zenith Total Delay (ZTD)

Although quality control has been implemented in the GPS data pre-processing and processing, a small number of low-quality and erroneous estimates are unavoidable. Therefore, we screened the GPS ZTD estimates and their formal errors
($\sigma_{ZTD}$) by employing a general range check and a station-specific outlier check (Bock et al., 2020). The range check is intended to exclude apparently unrealistic outliers. To determine a reasonable ZTD range, we calculated the global ZTD by using the 1-hourly 0.25°×0.25° ERA5 product and found that the ZTD at the Earth's surface was between 1096.5 and 2835.8 mm in 2020 (see Appendix A). We therefore set the ZTD range as from 1000 to 3000 mm, which is believed to be wide




enough to preserve all the natural variations over the world. In addition, the formal errors of ZTD ($\sigma_{ZTD}$) are a good indicator

for its quality. We rejected $\sigma_{ZTD}$ estimates larger than 6 mm. This threshold is appropriate for GPS ZTD solutions produced using GIPSY as reported by Bock et al. (2020). Three stations were discarded as their $\sigma_{ZTD}$ values were completely out of range. On average, the range checks of the ZTD and $\sigma_{ZTD}$ rejected 0.1% of the datapoints.

The ZTD and $\sigma_{ZTD}$ outlier checks were conducted with station-specific thresholds. For the $\sigma_{ZTD}$, an upper limit of $2 \times median(\sigma_{ZTD})$ was used as recommended by Bock et al. (2020), where $median(\sigma_{ZTD})$ is the median value of a given

station's $\sigma_{ZTD}$ within the whole year. Unlike $\sigma_{ZTD}$, the temporal variations of IWV can be very complicated. We therefore used a moving window outlier detection approach with the Interquartile Range (IQR) rule (Upton and Cook, 1996). For the 5-min ZTD series in each day, we determined the threshold as [Q1-3×IQR, Q3+3×IQR], where IQR=Q3-Q1. The Q1 and Q3 are the 25[th] and 75[th] percentiles of all the 5-min ZTD data in a 15-day window centred on this day, respectively. This approach is more robust compared to the traditional approach based on standard deviations of ZTD residuals from a curve

fitting. As an example, Figure 3 illustrates the data screening method at station HCRL. Outliers in the $\sigma_{ZTD}$ and ZTD of HCRL were successfully excluded. Overall, the outlier checks took four iterations on average for all the stations and removed 0.33% of the datapoints.

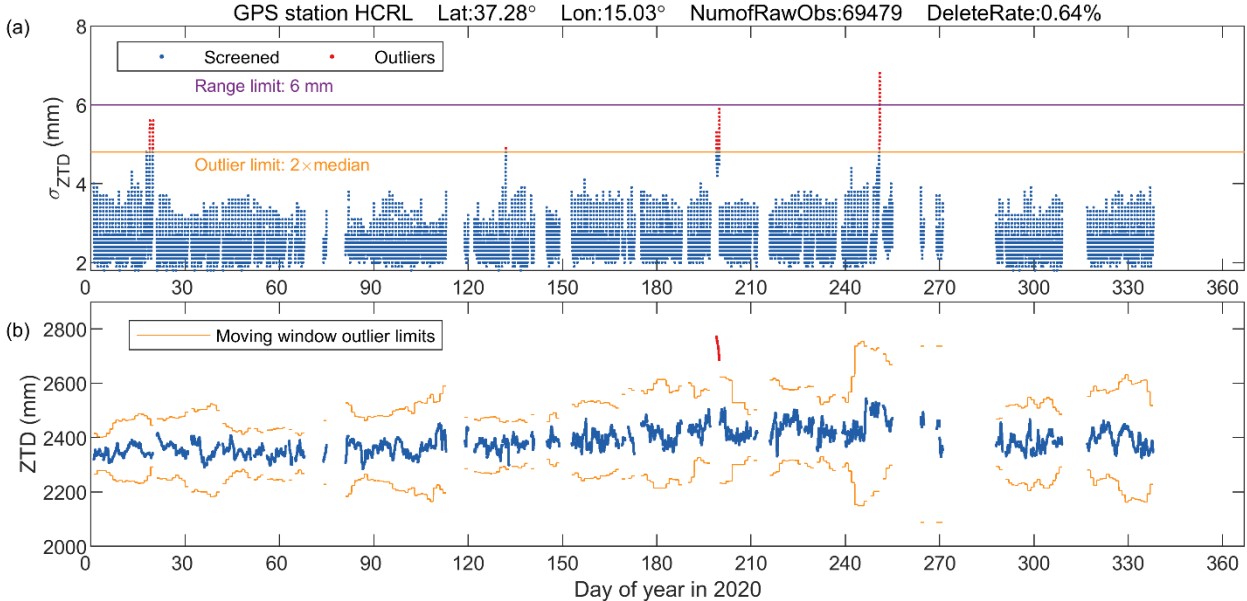

**Figure 3.** An example of the ZTD data screening at station HCRL. (a) Range check (purple line) and station-specific robust

outlier check (orange line) of $\sigma_{ZTD}$. (b) Robust outlier check (orange line) of ZTD with a 15-day moving window. The range check of ZTD was not shown because all the raw ZTD estimates are within the range (1000–3000 mm).

### 2.3 Retrieval approach of Integrated Water Vapour (IWV)

The screened GPS ZTD estimates were then converted into IWV by using the equations of (Bevis et al., 1992):



$$\text{IWV} = \Pi \cdot \text{ZWD}, \tag{1}$$

$$\Pi = \frac{10^6}{R_V \cdot [k_2' + k_3/T_m]}, \tag{2}$$

$$\text{ZWD} = \text{ZTD} - \text{ZHD}, \tag{3}$$

where $\Pi$ is the conversion factor (in kg m$^{-2}$ mm$^{-1}$) and $R_V = 461.522$ J kg$^{-1}$ K$^{-1}$ (Kestin et al., 1984) is the specific gas constant for water vapour. The parameters $k_2' = 22.1$ K hPa$^{-1}$ and $k_3 = 373900$ K$^2$ hPa$^{-1}$ are atmospheric refractivity constants (Bevis et al., 1994). $T_m$ denotes weighted mean temperature (in K), In the development of the enhanced GPS (enGPS) IWV product, we calculated $T_m$ at a given GPS station based on ERA5 pressure level product and the equation as follows:

$$T_m = \frac{\sum_{H_S}^{H_{top}} \frac{e_j}{T_j} \Delta H_j}{\sum_{H_S}^{H_{top}} \frac{e_j}{T_j^2} \Delta H_j}, \tag{4}$$

where $e$ and $T$ are water vapour pressure (in hPa) and temperature (in K) profiles from the geopotential altitude (i.e., geopotential height, $H_{gp}$) of the GPS station ($H_s$) to the top pressure level of ERA5 ($H_{top}$). Note that all the altitudes in this paper refer to $H_{gp}$ with unit of geopotential metre (gpm) unless expressly specified. The $H_{gp}$ is the altitude of a station above Mean Sea Level (MSL) by considering the variation of gravity. It is different from ellipsoidal and orthometric heights which are commonly used in GPS and Geodesy. The conversions of the height systems are provided in Appendix B. Moreover, in order to obtain the $T_m$ at the location of the GPS station, vertical adjustment (interpolation or extrapolation) and horizontal interpolation of associated meteorological variables from the gridded ERA5 product were carried out (see Appendix C).

The ZHD used in the enGPS IWV was calculated as follows (Saastamoinen, 1972; Davis et al., 1985):

$$\text{ZHD} = 2.2768 \frac{P_s}{1 - 2.66 \times 10^{-3} \cdot \cos(2\varphi_s) - 2.8 \times 10^{-7} H_s}, \tag{5}$$

where $P_s$ is GPS station pressure (in hPa) which is also estimated by vertical adjustment and horizontal interpolation (see Appendix C). $\varphi_s$ and $H_s$ are latitude (in rad) and orthometric height (in m) of the GPS station.

Eqs. (1)–(4) indicate that $T_m$ and ZHD (or $P_s$) are essential for the calculation of GPS IWV. The relative error of IWV due to an error of $\Delta T_m$ in $T_m$ can be estimated as follows:

$$\frac{\Delta \text{IWV}}{\text{IWV}} = \frac{\Delta \Pi}{\Pi} = \frac{(T_m + \Delta T_m)}{1 + r(T_m + \Delta T_m)} \cdot \frac{1 + rT_m}{T_m} - 1 \approx \frac{\Delta T_m}{T_m}, \tag{6}$$

where $r = k_2'/k_3 = 5.9 \times 10^{-5}$ K$^{-1}$ is close to zero, and hence the relative error of IWV and $T_m$ are approximately identical (Wang et al., 2005). Accordingly, 1 K of $T_m$ error can result in a relative error of 0.33–0.48% in IWV, given that the ERA5-derived global surface $T_m$ ranging between 208.9 and 304.9 K in 2020 (see Appendix A). The ZHD error is proportional to the associated IWV error with the factor of $-\Pi$, which can be calculated from $T_m$. For the range of global $T_m$ in 2020, the value of $\Pi$ ranges between 0.12 and 0.17 kg m$^{-2}$ mm$^{-1}$.

We also downloaded NGL's operational GPS (opGPS) IWV product for a comparison. Unlike the enGPS IWV product, the ZHD and $T_m$ used in the opGPS IWV were obtained from Vienna University of Technology (TUW,



https://vmf.geo.tuwien.ac.at/) during the GPS data processing. TUW calculates the ZHD and $T_m$ 6-hourly at the Earth's

surface with a 2°×2.5° grid based on an operational ECMWF product (Boehm et al., 2006). Although vertical adjustments of

both ZHD and $T_m$ from the ECMWF surface level to the altitude of GPS station are essential for the accurate calculation of

GPS IWV, only ZHD was vertically adjusted in NGL's opGPS IWV product as described in Appendix D. The vertical

change of $T_m$ has not been considered in the opGPS IWV product because an accurate lapse rate of $T_m$ is hard to obtain.

   The latest ERA5 atmospheric reanalysis is expected to provide more accurate $P_s$ (or ZHD) and $T_m$ than the TUW

product, although it has a time lag from five days to three months. This is because ERA5 has a much higher spatiotemporal

resolution, with 1-hourly estimates at 0.25°×0.25° horizontal grids and 37 vertical pressure levels. Moreover, atmospheric

reanalysis products are generally considered to be superior to the operational analysis products due to improvements in input

data and assimilation algorithms (Dee et al., 2009). Details on the extraction of $P_s$ and $T_m$ at the location of a given GPS

station from the 1-hourly ERA5 pressure level product are provided in Appendix C. The meteorological parameters were

then linearly interpolated into 5-min time series and used for the conversion of enGPS IWV from NGL's ZTD.

## 2.4 Screening of IWV

   The 5-min enGPS IWV data were screened with a range check and a moving window outlier check. We first

determined the IWV range by inspecting global IWV at the Earth's surface derived by ERA5 (see Appendix A). Results

showed that the global IWV in 2020 ranges from 0.0 to 94.9 kg m$^{-2}$ (Fig. A1a-b). We therefore set the range of the check as

0–100 kg m$^{-2}$. The range check showed that no datapoint in the enGPS IWV product has a value larger than 100 kg m$^{-2}$.

However, 0.03% (292,143) datapoints showed values less than zero, which could be due to inaccurate ZTD -or ZHD

estimates. The physically unrealistic estimates were mostly found at high-latitude stations with complex terrains, such as

Antarctica and Greenland. These sites have low humidities and might have large spatial interpolation errors in ZHD.

Moreover, the moving window outlier check rejected 0.16% of the IWV datapoints.

In the end, we obtained 1,093,591,492 5-min enGPS IWV estimates at 12,552 stations in 2020, which is equal to

87,125 datapoints per station and 10,375 datapoints every 5-min. The average completeness rate was 82.7%, and the rates

were higher than 95% at 7,253 stations.

   For a fair comparison, we also screened the opGPS IWV product before evaluating its quality. The range check

excludes 0.35% (3,849,630) of the datapoints with unrealistic negative IWV estimates, which is 13 times larger than that of

enGPS. In particular, the IWV at four stations (CON2, MCG2, NAU2, and PHIG) on the high mountains (>3000 gpm) of

Ross Island, Antarctica, were entirely smaller than zero in opGPS but rarely negative in enGPS. As an example, Figure 4

compares their IWV estimates at station PHIG (77.53°S, 167.05°E, 3185.9 gpm). All the opGPS IWV estimates are

unrealistically below zero, with values ranging from -13.0 to -1.7 kg m$^{-2}$. In contrast, the enGPS IWV are between -0.3 and

2.5 kg m$^{-2}$ with negative values at only 0.32% (335 out of 105,389) of the datapoints. The negative opGPS IWV estimates at

station PHIG are likely caused by the inaccurate ZHD provided by TUW with low spatial resolution and an unrealistic



approximation of the barometric formula (Eq. (D1) in Appendix D) used to adjust pressure from the TUW model surface to an extremely high altitude.

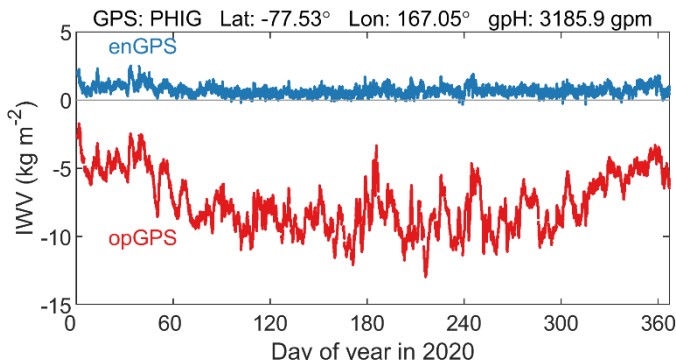

**Figure 4.** Comparison of the unscreened enGPS (blue) and opGPS (red) IWV at station PHIG (77.53°S, 167.05°E, 3185.9
gpm). $H_{gp}$ is geopotential altitude in geopotential metre (gpm; see Appendix B).

**2.5 In-situ meteorological observations**

Intercomparison of IWV products derived by different techniques is crucial for the evaluations of their consistency and quality. In this work, we aggregated the 5-min GPS IWV and associated $T_m$ and ZHD data into 1-hourly time series with a threshold of at least four 5-min datapoints in each moving 1-hour time window centred at a full hour. We then compared the
1-hourly enGPS and opGPS IWV products and their $T_m$ values with those of in-situ radiosonde (RS) observations collected by Integrated Global Radiosonde Archive (IGRA) version 2 (Durre et al., 2018). RS measurements have traditionally been used as validation sources for other humidity measurements (e.g., Lindstrot et al., 2014). IGRA is one of the most comprehensive RS datasets consisting of over 2,700 stations worldwide, of which about 800 are currently operational with regular daily observations at 00 and 12 UTC. We used temperature and relative humidity profiles from the IGRA dataset and
carried out quality control with the following criteria: 1) the profiles reach the surface and a pressure level of at least 300 hPa on top (Wang and Zhang, 2008); 2) data are available at no less than five (four) standard pressure levels for stations below (above) the 1000-hPa pressure level (Wang et al., 2005); 3) pressure gaps should be less than 200 hPa.

In addition to $T_m$, the quality of ZHD also has an influence on GPS IWV retrievals. We employed the ZHD calculated by using Synoptic (SYN) observations from Integrated Surface Database (ISD; Smith et al., 2011) as references to evaluate
the ZHD values estimated from ERA5 and TUW. ISD has more than 20,000 SYN stations worldwide and provides high-quality observations (e.g., pressure) with internal quality control procedures.



## 3 Characterisations of the GPS IWV product

We analysed the statistical characteristics of each 5-min enGPS IWV time series by calculating the minimum, maximum, and mean value as well as the coefficient of variation (CV), which is the ratio of the SD to the mean. To avoid
impacts of missing data on the estimates, we selected 9,418 out of the 12,552 stations with a criterion that their completeness rates were higher than 50%, and their data gaps were shorter than 30 days. The IWV time series of the selected stations range from 0 to 82.7 kg m$^{-2}$ (Fig. 5a–b). The geographical patterns of the minima and maxima of the GPS IWV are consistent with those from ERA5 (Fig. A1a–b). The minima of the GPS IWV generally increase towards the equator with values close to zero at the poles (66.5°N–90°N or 66.5°S–90°S) and median values of 1.5 and 9.1 kg m$^{-2}$ in mid-latitude (23.5°N–66.5°N or
23.5°S–66.5°S) and tropical (23.5°N–23.5°S) zones, respectively. However, the geographical characteristics of the maxima IWV are more complicated, as they depend on climate type and altitude in addition to latitude. The lower maxima were observed at the polar climate zones or high-altitude regions with the lowest value of 2.2 kg m$^{-2}$ at station NAU2 (77.52°S, 167.15°E, 3632 gpm) on Ross Island, Antarctica (Fig. 5e). Moreover, the maxima of most of the stations in Antarctica are generally lower than 10 kg m$^{-2}$ except for the Antarctic Peninsula. This is not surprising because Antarctica is known as the
coldest, driest, and highest continent on Earth. In contrast, the monsoon zones or low-altitude regions are characterised with larger maxima by the largest value of 82.7 kg m$^{-2}$ at station J731 (28.37°N, 130.03°E, 11 gpm) on Ryukyu Islands, Japan, during the Typhoon Haishen on Sept. 6, 2020.

The mean values of the stations' IWV estimates generally increase towards the equator, with the lowest and highest values of 0.6 and 55.5 kg m$^{-2}$ found at station NAU2 in Antarctica and ANMG (2.8°N, 101.51°E, 18 gpm) in Malaysia,
respectively. As can be seen in Fig. 5e, the variation of IWV at station ANMG is rather small compared to its mean value, and hence, it has one of the lowest CV with a value of 9.1% (Fig. 5d). In contrast, the largest CV with a value of 100% is observed at station REPL (66.52°N, 86.23°W, 21 gpm) at Repulse Bay, Canada, where both the mean and SD of IWV are equal to 6.9 kg m$^{-2}$ (Fig. 5e).





**Figure 5.** Minima (a), maxima (b), mean values (c), and coefficients of variations (d) of the 5-min enhanced GPS IWV time series in 2020 at 9,418 stations. (e) IWV time series of four exemplary stations.

## 4 Evaluation with in-situ observations

We selected 182 GPS-RS-SYN station clusters by using a spatial and temporal matching approach with the following criteria: 1) the distances of the RS and SYN stations with respect to their paired GPS stations are within 50 km in horizontal and 100 m in vertical (Wang and Zhang, 2008); 2) The time-matched GPS, RS and SYN observations are available at more than 50% of the 00 and 12 UTC epochs in 2020 and all the gaps in time are shorter than 30 days. The locations of the station clusters are listed in Table S1. We conducted vertical adjustments of humidity, temperature, and pressure observed at the RS to the altitude of their paired GPS station and then calculated the associated IWV and $T_m$, We also adjusted the SYN





pressure measurements vertically to the altitude of their paired GPS station for the calculation of referenced ZHD. Details of
the vertical adjustments of RS and SYN observations are provided in Appendix C. After that, we screened the IWV
differences by employing a 3-fold IQR rule with a 90-day moving window. Similarly, the $T_m$ differences and ZHD
differences were also screened.

Figure 6a shows the mean RS-enGPS IWV differences at the 182 station clusters. The values are within ±3 kg m$^{-2}$.
North America is characterised by a general negative mean IWV difference, indicating a dry bias of RS with respect to
enGPS or a wet bias of enGPS with respect to RS. In contrast, the differences are positive at a part of European stations,
indicating a wet bias of RS or a dry bias of enGPS. The IWV biases could be due to systematic errors and technique-related
differences in RS and GPS as well as inaccurate meteorological information. We expand a little bit further on this.

As humidity biases in specific RS types are well-known (e.g., Wang and Zhang., 2008), they are investigated first. We
classified the RS stations into seven subsets (namely Graw, LMS6, M10, Meisei, RS41, RS92, and "Others") according to
their manufacturers and sensor types in 2020 (Fig. 1). The subset of "Others" includes stations with unknown types or type
changes. Figure 7 shows the IWV difference between RS and enGPS for each subset. Good agreements are observed in the
subsets of Meisei, RS41, and RS92 with mean IWV differences of -0.4, 0.3, and 0.5 kg m$^{-2}$, respectively. However, the Graw
and LMS6 stations, mostly located in North America and the Pacific, are generally characterised by noticeable negative
mean RS-enGPS IWV differences down to -3.0 with averages of -0.9 and -1.1 kg m$^{-2}$, respectively. In contrast, the M10
stations, all located in Europe, obtain positive mean RS-enGPS IWV differences ranging from 0.5 to 1.3 with an average of
1.0 kg m$^{-2}$.





**Figure 6.** Comparisons of IWV from radiosonde observations and those from enGPS (left) or opGPS (right) IWV retrievals
at 182 station clusters in mean value (a and b) and standard deviation (c and d) of their differences (radiosonde minus GPS),
and Pearson correlation coefficient (e and f). The enGPS and opGPS indicate the enhanced GPS IWV product developed in
this work and the operational GPS IWV product provided by NGL, respectively. The significant (squares) and insignificant
(circles) mean values and correlation coefficients are derived from tests at 95% confidence level.

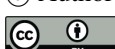


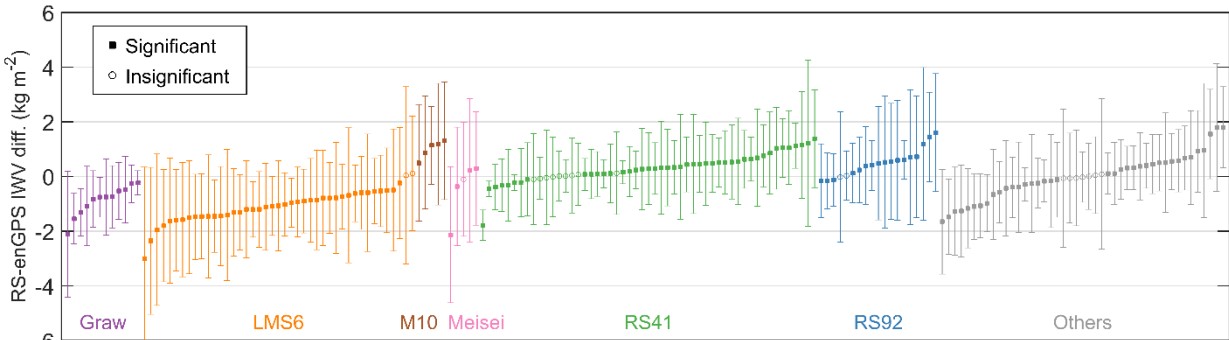

**Figure 7.** Mean values and standard deviations of the IWV differences (radiosonde minus enGPS) at the 182 station clusters. The colours of the bars indicate different radiosonde types

**Table 1.** Mean IWV differences between RS and enGPS at daytime, nighttime, and in whole day. The daytime and nighttime were determined as observations of stations within 66.5°S–66.5°N at local time of 8am–6pm and 8pm–6am, respectively. Accordingly, stations at 90°E–120°E and 60°W–90°W were excluded because their local time is 6am–8am (dawn) or 6pm–8pm (twilight) at 00 and 12 UTC, at which the RS observations are available. In the end, 131 out of the 182 RS-enGPS station clusters were selected for the daytime and nighttime comparison.

| RS type | Number of station clusters | Mean RS-enGPS IWV difference ($kg\ m^{-2}$) | | |
|---|---|---|---|---|
| | | Daytime | Nighttime | Whole day |
| Graw | 5 | -1.1 | -0.9 | -1.0 |
| LMS6 | 26 | -1.5 | -0.7 | -1.1 |
| M10 | 5 | 0.8 | 1.1 | 1.0 |
| Meisei | 5 | -1.4 | 0.5 | -0.4 |
| RS41 | 42 | 0.3 | 0.3 | 0.3 |
| RS92 | 9 | -0.1 | 0.7 | 0.3 |
| Others | 39 | -0.2 | 0.1 | 0.0 |

As the RS humidity measurements could be impacted by solar radiation bias, we further evaluated and compared the IWV biases of RS with respect to GPS at daytime and nighttime. The daytime and nighttime of a RS station were determined as its local time of 8am–6pm and 8pm–6am, respectively. Stations with their observations available at local time of 6am–8am (dawn) or 6pm–8pm (twilight) were excluded. Stations in polar regions (66.5°S –90°S or 66.5°N –90°N) were also removed. In the end, there are 131 out of the 182 RS-enGPS station clusters selected for the daytime and nighttime comparison. Table 1 lists the mean RS-enGPS IWV difference at daytime, nighttime, and in whole day for each RS type. Both mean RS41-enGPS IWV differences at daytime and nighttime are identical to a small value of 0.3 $kg\ m^{-2}$, indicating a good agreement between RS41 and GPS in both scenarios. However, although the mean Meisei-enGPS IWV difference is weak in whole day (-0.4 $kg\ m^{-2}$), it is an average of the significant negative value at daytime (-1.4 $kg\ m^{-2}$) and the moderate positive value at nighttime (0.5 $kg\ m^{-2}$), respectively. Likewise, the mean RS92-enGPS IWV differences are -0.1, 0.7, and 0.3 $kg\ m^{-2}$ at daytime, at nighttime, and in whole day, respectively. The results suggest that it is important to evaluate the





mean RS-enGPS IWV difference at daytime and nighttime separately, even though the mean difference in whole day is weak. Moreover, both the mean LMS6-enGPS IWV differences at daytime and nighttime are negative with values of -1.5 and -0.7 kg m$^{-2}$, respectively. The results indicate a stronger dry bias of LMS6 with respect to enGPS at daytime than nighttime. Similar results are seen in Graw. The mean Graw-enGPS IWV differences are -1.1 and -0.9 kg m$^{-2}$ at daytime and nighttime, respectively. By contrast, the mean M10-enGPS IWV differences are positive at daytime and nighttime with values of 0.8 and 1.1 kg m$^{-2}$, respectively. In addition, we also investigated the GPS receiver and antenna types of these station clusters, but we did not find obvious relationships with the IWV differences. The results suggest that the mean RS-enGPS IWV differences might be more related to the RS types rather than the GPS types, but other factors like inaccurate meteorological information cannot be excluded.

Inaccurate meteorological information could deteriorate GPS IWV retrievals. Section 2.3 have indicated that the relative error of $T_m$ and the associated relative error of IWV are approximately equal, while 1 mm of ZHD bias can result in a negative IWV bias ranging from -0.12 to -0.17 kg m$^{-2}$. In order to find out why the mean RS-enGPS IWV difference for the Graw, LMS6, and M10 are more significant than the other types, we compared the ERA5-derived $T_m$ and ZHD with reference values from the in-situ RS and SYN observations, respectively. The mean RS-ERA5 $T_m$ differences are 0.2–2.3 K for the 55 Graw and LMS6 station clusters (Fig. 8a), equivalent to relative $T_m$ differences of 0.1%–0.8%, whereas the associated relative RS-enGPS IWV differences are -17.2%–0.3% with an average of -6.5%. Likewise, the relative RS-ERA5 $T_m$ differences for the five M10 station clusters are 0–0.2%, whereas the associated relative RS-enGPS IWV differences are 2.6%–6.3% with an average of 4.7%. Regarding the ZHD, the mean SYN-ERA5 difference is -0.7 mm for the Graw and LMS6 station clusters (Fig. 9a), which would only result in an average IWV bias of 0.1 kg m$^{-2}$, but the actual value is -1.0 kg m$^{-2}$. As for the M10 station clusters, their mean SYN-ERA5 ZHD difference is -0.6 mm, suggesting a mean RS-enGPS IWV difference inferior to 0.1 kg m$^{-2}$, which is an order of magnitude lower than the actual value of 1.0 kg m$^{-2}$. The results indicate that the errors of the ERA5-derived $T_m$ and ZHD are very unlikely to be the primary reason for the significant RS-enGPS IWV for these three RS types. Although it seems that the systematic differences in RS-enGPS IWV are related to the specific RS types, we cannot exclude other factors, such as GPS data processing models and strategies. Intercomparisons with other water vapour retrieval techniques will contribute to a more comprehensive understanding of the disagreements, but it is beyond the scope of this work and will be addressed in the future.







**Figure 8.** Comparisons of $T_m$ from radiosonde observations and those from ERA5 (left) or TUW at 182 station clusters in mean value (a and b) and standard deviation (c and d) of their differences (radiosonde minus ERA5 or TUW), and Pearson correlation coefficient (e and f). The $T_m$ obtained from ERA5 and TUW were used in the retrievals of enGPS and opGPS IWV products, respectively.





**Figure 9.** Comparisons of ZHD from synoptic (SYN) pressure observations and those from ERA5 (left) or TUW at 182 station clusters in mean value (a and b) and standard deviation (c and d) of their differences (SYN minus ERA5 or TUW), and Pearson correlation coefficient (e and f). The ZHD obtained from ERA5 and TUW were used in the retrievals of enGPS and opGPS IWV products, respectively.

## 5 Evaluation of enGPS versus opGPS

The enGPS IWV product is different from opGPS in the fact that the conversion of IWV from ZTD was carried out by using the high spatiotemporal resolution reanalysis ERA5 instead of TUW, and thus a quality improvement is expected. We assess the quality of both GPS IWV datasets by taking the IWV from RS as reference. We are aware that the IWV measured by specific RS types could be biased, and thus we will interpret the results with caution by considering differences in the meteorological data for GPS IWV retrieval and the topographical complexity of terrains around the stations.

Figure 10 compares the IWV consistency in mean value and SD of RS-GPS differences as well as linear Pearson correlation coefficient at each station cluster. The majority of the mean RS-GPS IWV differences are within ±3 kg m$^{-2}$ for both enGPS and opGPS (Fig. 10a). However, the Mean Absolute Bias (MAB) decreases from 0.86 to 0.69 kg m$^{-2}$ by the development of enGPS, equal to a reduction of 19.5%. In particular, the mean IWV differences in RS-opGPS turn out to be negligible in RS-enGPS at several stations, e.g., stations NAVI, PAVI, and J500 with values from 8.0, 2.5, and -2.8, to 0.3,

0.3, and 0.2 kg m$^{-2}$, respectively. Their improvements will be carefully analysed in latter part of this section. In addition, enGPS IWV has a better agreement with RS in variability. Compared to opGPS, the average SD of IWV differences decreases from 1.58 to 1.48 kg m$^{-2}$, indicating a reduction of 6.2% (Fig. 10b). The most significant SD reduction can be found at station PAVI, with its SD decreasing from 2.5 to 1.4 kg m$^{-2}$. Also, enGPS IWV achieves a slightly stronger correlation with respect to RS than that between opGPS and RS, with average correlation coefficients of 0.986 and 0.983,

respectively. The correlation improvement is more significant at specific stations (Fig. 10c). For example, the correlation coefficient is increased from 0.89 to 0.94 at station NAVI. Inspecting the maps of IWV comparisons shown in Fig. 6, we find that the improvements in the IWV agreements are most significant in regions with a considerable topographical complexity of terrains, such as the Alps, the Rocky Mountains, the Andes, and Antarctic coast. The results indicate that enGPS IWV is superior to opGPS.

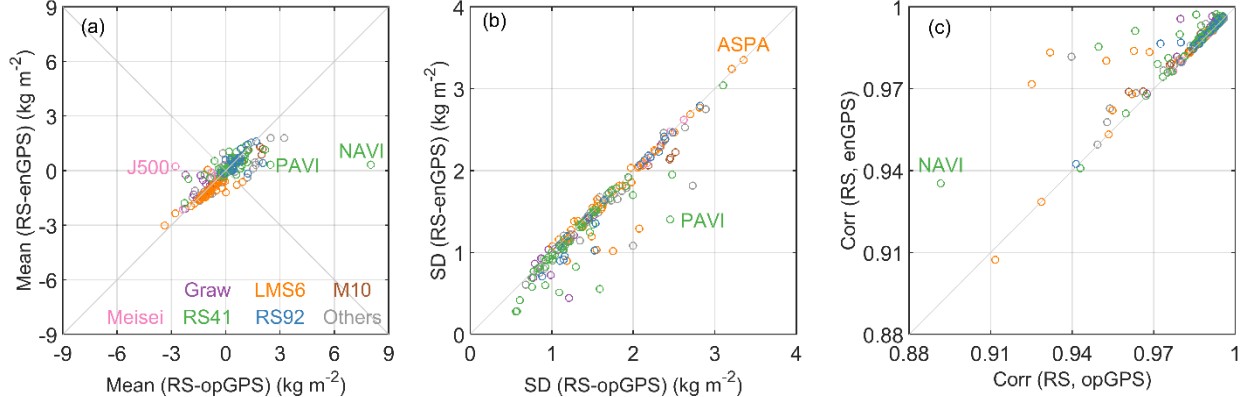


**Figure 10.** Comparisons of the mean values (a), and standard deviations (b) of the IWV differences between radiosonde and opGPS or enGPS, and correlations (c) between IWV from radiosonde and opGPS or enGPS. The colours of the circles indicate different radiosonde types.



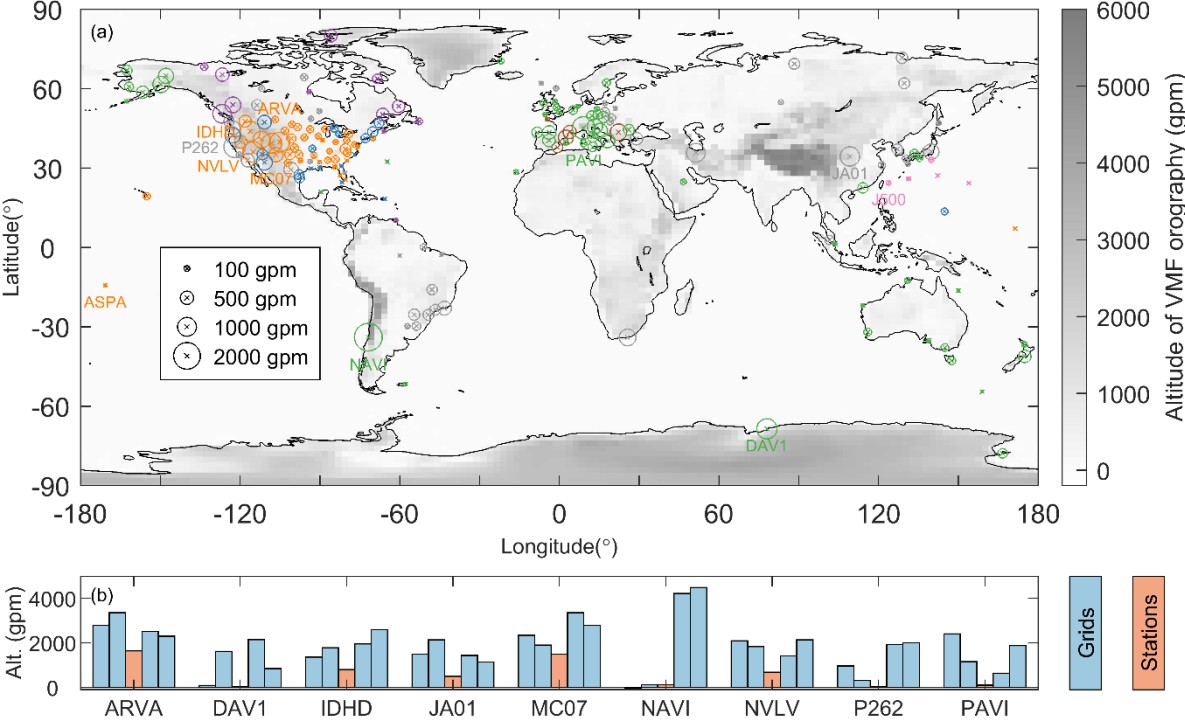

**Figure 11.** (a) Mean absolute altitude difference ($\overline{\Delta H}$) between the altitudes of each GPS station and its four surrounding TUW grid nodes indicated by the sizes of circles. The colours of the circles indicate the radiosonde types as shown in Fig. 1. The grey-scale grids are the geopotential altitudes of the TUW model surface converted from ellipsoid heights (see Appendix B). (b) Altitudes of the nine GPS stations with their $\overline{\Delta H}$ larger than 1000 gpm. The orange and blue bars indicate the altitudes of GPS stations and their specific four surrounding TUW grid nodes, respectively.

To explain the improvements in IWV agreements, we evaluated the meteorological information from ERA5 and TUW by taking RS-derived $T_m$ and SYN-derived ZHD as references, respectively. Figures 8–9 show that the ERA5-derived $T_m$ and ZHD outperform those from TUW with the most striking improvements also in regions with complex topographical terrains. The results indicate that the enhancements of ERA5 in spatial resolution (0.25°×0.25°, 37 levels) might enable to resolve local variations of the meteorological variables in the complex topographical areas. To evaluate the impacts of complex terrains on the performance of the TUW product, we calculated the mean absolute altitude difference ($\overline{\Delta H}$) for each GPS station:

$$\overline{\Delta H} = \text{mean}(|H_i - H_s|), \tag{7}$$

where $H_s$ and $H_i$ ($i = 1, 2, 3, 4$) are the altitudes of the GPS station and its four adjacent TUW grid nodes, respectively. The $\overline{\Delta H}$ is considered as a topographical complexity index as displayed in Fig. 11a. When the $\overline{\Delta H}$ is larger, the spatial representativeness differences between the in-situ meteorological observations and the gridded TUW products tends to be

more significant. It is seen from Fig. 11a that the $\overline{\Delta H}$ values are larger in the Alps and the Rocky Mountains. In addition, extreme values of $\overline{\Delta H}$ can also be observed at the Andes and Antarctic coast, despite only a few stations there. Figure 11b illustrates the altitudes of nine stations whose $\overline{\Delta H}$ are larger than 1000 gpm. The largest value of $\overline{\Delta H}$ is found at station NAVI with a value of 2160 gpm.

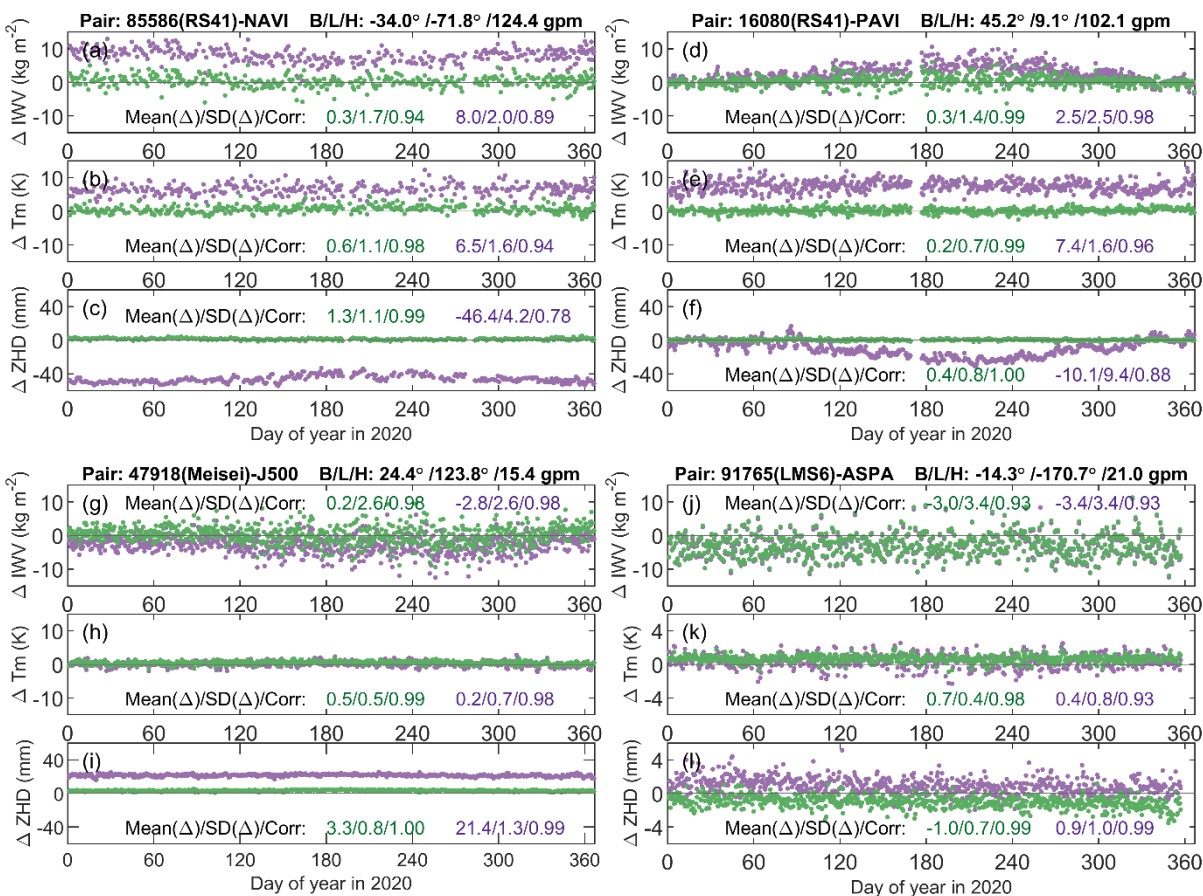

**Figure 12.** (a) IWV differences between radiosonde (RS) observations and enGPS (green) or opGPS (purple) at station cluster 85586-NAVI. (b) $T_m$ differences between radiosonde and ERA5 for enGPS (green) or TUW for opGPS (purple). (c) ZHD differences between synoptic (SYN) observations and ERA5 (green) or TUW (purple). (d,g,j), (e,h,k), and (f,i,l) are the same as (a), (b), and (c) but for the other three station clusters, respectively. Note that the ranges of Y-axes in (k) and (l) are different from the others.

Station NAVI (34.0°S, 71.8°W, 124.4 gpm) is located at a coastal area of Chile between the Pacific and the rugged Andes. Two of its four adjacent grid nodes are close to mean sea level, while the other two are higher than 4200 gpm. We analysed the impacts of spatial representativeness difference due to the strong variations of topographical relief on the estimates of its IWV as well as associated $T_m$ and ZHD (Fig.12a–c). The mean IWV difference reaches 8.0 kg m$^{-2}$ at NAVI for RS-opGPS but reduced to 0.3 kg m$^{-2}$ for RS-enGPS. The results indicate relative mean IWV differences of 58% and 2%,





respectively, by referring to its RS-derived IWV with a mean value of 13.8 kg m$^{-2}$. As for the associated $T_m$, the mean differences in RS-ERA5 and RS-TUW are 0.6 and 6.5 K (i.e., 0.2% and 2.3%), respectively. Moreover, the mean ZHD differences in SYN-ERA5 and SYN-TUW are 1.3 and -46.4 mm, respectively. As we know that the relative bias of $T_m$ results in nearly the same amount of relative bias of IWV, the IWV biases due to $T_m$ should be only 0.0 and 0.3 kg m$^{-2}$ for

RS-enGPS and RS-opGPS, respectively. In contrast, the IWV biases due to ZHD are -0.2 and 7.4 kg m$^{-2}$ in RS-enGPS and RS-opGPS, respectively, by assuming $\Pi = 0.17$ in Eq. (1). Therefore, the considerable mean SYN-TUW ZHD difference at station NAVI is likely to be the primary reason for its large mean RS-opGPS IWV difference. Similar results are also observed at many other stations situated in regions with complex topography, such as the Alps, the Rocky Mountains, and the Antarctic coast (see e.g., Fig. 6-8), The results indicate that the improvement of ERA5 in spatial representativeness with

respect to TUW enables a significant mitigation of mean IWV difference in RS-enGPS with respect to RS-opGPS.

The RS-opGPS IWV difference can also be dependent on season. For example, it can be as large as 10.5 kg m$^{-2}$ in summer whereas mostly within ±2 kg m$^{-2}$ in winter at station PAVI (45.2°N, 9.1°E, 102.1 gpm; Fig. 12d), which is in northern Italy at the foothills of the Alps. Similarly, its SYN-TUW ZHD difference time series is also characterized with a strong seasonal pattern with a SD of 9.4 mm (Fig. 12f). As the $\overline{\Delta H}$ of station PAVI is quite large (1418 gpm), its seasonal-

dependent RS-opGPS IWV difference can be attributed to the strong spatial representativeness difference of the gridded TUW-derived ZHD with respect to that from the in-situ SYN. By contrast, the seasonal variation of the SYN-ERA5 ZHD difference is quite weak, with a SD of only 0.8 mm. As a result, the SD of IWV difference is reduced from 2.5 in RS-TUW to 1.4 kg m$^{-2}$ in RS-ERA5 for this station. The results indicate that ERA5 has a better spatial representativeness of seasonal variations of ZHD than TUW at station PAVI, and thus contributes to more accurate GPS IWV retrievals over time.

In addition, the benefit of the enhanced spatial representativeness of ERA5 with respect to TUW is also significant for stations on islands and in coastal regions. For instance, the mean ZHD difference is reduced from 21.4 mm in SYN-TUW to 3.3 mm in SYN-ERA5 at station J500 (24.4°N, 123.8°E, 15.4 gpm) on an island of Okinawa, Japan (Fig. 12i). Consequently, the mean IWV difference has been substantially reduced from -2.8 kg m$^{-2}$ in RS-opGPS to 0.2 kg m$^{-2}$ in RS-enGPS (Fig. 12g). The more significant mean IWV difference in RS-opGPS could be due to the noticeable SYN-TUW ZHD differences

with an average of 21.4 mm (Fig. 12i), which could result in an IWV bias in the range from -3.6 to -2.6 kg m$^{-2}$.

There are also stations at which the IWV differences cannot be properly explained by the spatial representativeness differences of $T_m$ and ZHD. For example, at station ASPA (24.4°S, 123.8°W, 15.4 gpm) on an island of Samoa, the mean IWV differences in RS-enGPS and RS-opGPS reach -3.0 and -3.4 kg m$^{-2}$, respectively. Since the average IWV from RS is 47.9 kg m$^{-2}$ at this station, the mean $T_m$ differences in RS-ERA5 and RS-TUW, with values of 0.7 and 0.4 K (i.e., 0.2% and

0.1%), would only lead to mean IWV differences inferior to 0.1 kg m$^{-2}$ in both RS-enGPS and RS-opGPS. As for the mean ZHD differences in SYN-ERA5 and SYN-TUW with values of -1.0 and 0.9 mm, the corresponding absolute mean IWV differences in RS-enGPS and RS-opGPS should be less than 0.2 kg m$^{-2}$. Hence, the spatial representativeness differences in $T_m$ and ZHD are very unlikely to result in such significant IWV differences. Moreover, the distance between the station cluster of 91765-ASPA is only 1.1 km in horizontal and 17 gpm in vertical, indicating a weak representativeness difference





between the GPS- and RS-derived IWV. A more likely reason could be the systematic dry bias of the LMS6 sensor as
analysed in Section 4. Furthermore, the SD values of the RS-enGPS and RS-opGPS IWV differences are identical at station
ASPA, with a value of 3.4 kg m$^{-2}$ (Fig. 12j). This is the maxima SD value over all the 182 station clusters in both enGPS and
opGPS IWV datasets (Fig. 10b), indicating that the poor agreement could also be due to station-specific errors in the GPS or
RS measurements. Further comparisons with independent IWV measurements derived by other techniques will be helpful to

explain the poor agreement, and it will be addressed in future studies.

**6 Summary and outlook**

A global dataset of high-quality ground-based GPS Integrated Water Vapour (IWV) has been developed in this work. It
consists of 1,093,591,492 IWV estimates at 12,552 GPS stations worldwide in 2020 with a temporal resolution of 5-min,
which is equal to 87,125 datapoints per station or 10,375 datapoints every 5 minutes. The dataset is derived by using precise

GPS tropospheric Zenith Total Delays (ZTD) provided by Nevada Geodetic Laboratory (NGL) as well as Zenith Hydrostatic
Delays (ZHD) and weighted mean temperatures ($T_m$) obtained from the newly released ERA5 atmospheric reanalysis with a
very high spatiotemporal resolution (0.25°×0.25°, 37 vertical levels, 1-hourly). A dedicated robust data screening approach
is also implemented in order to avoid the impacts of outliers. The dataset can be considered as an enhanced version of NGL's
IWV product compared to its operational version, which is produced routinely by using the meteorological information from

TU Wien (TUW) with a much coarser resolution (2°×2.5°, surface level, 6-hourly).

The global GPS IWV values range between 0.0 and 82.7 kg m$^{-2}$ in 2020. Its stochastic characteristics can be related to
many factors, such as latitude, altitude, topography, and climate type. For instance, the maxima of IWV are mostly lower
than 10 kg m$^{-2}$ in Antarctica with high altitude as well as cold and dry climate condition. The average IWV of the stations
generally increase from the poles towards the equator with values from 0.6 to 55.5 kg m$^{-2}$.

The enhanced GPS (enGPS) IWV product has a good agreement with global radiosonde observations. The Mean
Absolute Bias (MAB) at 182 collocated station clusters is only 0.69 kg m$^{-2}$. However, systematic effects related to specific
radiosonde types are noticed. Although the mean RS-enGPS IWV differences are within ±0.5 kg m$^{-2}$ for types of Meisei,
RS41, and RS92, they are more significant for Graw, LMS6, and M10, with values of -0.9, -1.1, and 1.0 kg m$^{-2}$, respectively.
In addition, the mean RS-enGPS differences at daytime and nighttime can be significantly different in specific RS types,

although they are identical to 0.3 kg m$^{-2}$ in RS41-enGPS IWV. For example, the mean Meisei-enGPS IWV difference at
daytime and nighttime are -1.4 and 0.5 kg m$^{-2}$, respectively. The values for RS92-enGPS IWV difference are -0.1 and 0.7 kg
m$^{-2}$, respectively. Both the mean LMS6-enGPS IWV differences at daytime and nighttime are negative with values of -1.5
and -0.7 kg m$^{-2}$, respectively. Similarly, the values for Graw-enGPS IWV difference are -1.1 and -0.9 kg m$^{-2}$, respectively.
By contrast, both the mean LMS6-enGPS IWV differences at daytime and nighttime are positive with values of 0.8 and 1.1

kg m$^{-2}$, respectively. The significant IWV differences in these three radiosonde types are more likely to be related to their



own systematic effects (e.g., solar radiation effect) rather than GPS device types or meteorological data, but other factors cannot be excluded.

The enGPS IWV is generally superior to NGL's operational GPS (opGPS) IWV product. With the development of enGPS IWV, the mean and standard deviations of differences with respect to radiosonde-derived IWV are reduced by 19.5%
and 6.2% on average, respectively. The improvements are more obvious in regions with complex topographical terrains like the Alps, the Rocky Mountains, the Andes, and Antarctic coast. In addition, the number of unrealistic negative IWV estimates (3,849,630) in NGL's opGPS product has also been heavily reduced by 92.4%. These improvements are achieved because the high resolution ERA5 can capture small-scale variations of the meteorological variables in the complex topographical areas, which cannot be resolved by the TUW products due to its limited resolution. The results highlight the
importance of using the enGPS IWV product instead of opGPS for high-accuracy applications, such as the evaluation of subtle difference with respect to other IWV retrieval techniques. Moreover, the enGPS IWV product can be used for retrospective analysis of extreme weather events, although it is not available in real-time due to the time lags of related high-quality products for GPS data processing and high-resolution ERA5 for the conversion of IWV. Owing to the high temporal resolution, the enGPS IWV can contribute to the characterization of diurnal variation of IWV and analysis of related
hydrometeorological phenomena. With more than one billion of high-quality GPS IWV estimates and associated meteorological parameters provided, the dataset is also valuable for the development of related machine learning algorithms, such as predicting IWV in real-time from GPS ZTD.

Although the enhanced GPS IWV product already has more than 10,000 stations, it includes at this stage only the data available for 2020 and just sparse stations in regions like Africa, China, and India. The reason for the only one-year time
coverage is that the global 1-hour ERA5 pressure level product is huge (about 2 TB/year) and time-consuming for downloading and processing. The sparse spatial distribution of the GPS stations in specific regions is partly due to limitations in data sharing policies. Nevertheless, we plan to update this dataset regularly and extend it back to 1994. We will also include more GPS stations to achieve a better spatial coverage over the world's continents.

Finally, it is noted that the enhanced conversion of GPS-estimated ZTD to IWV does not affect GPS position estimates.
If higher resolution numerical models were implemented at the GPS data processing stage, then that should result in better position estimates together with the simultaneously estimated ZTD. Hence there will be a second-order enhancement of the IWV product. To do this goes beyond the scope of this work, but we suggest it as a promising extension of our investigation.

*Data availability.* The GPS ZTD data were provided by NGL (http://geodesy.unr.edu). The auxiliary meteorological data for
the retrieval of enGPS and opGPS IWV were derived by ERA5 from the Climate Data Store (CDS, https://cds.climate.copernicus.eu) and GRID map product of Vienna University of Technology (TUW, https://vmf.geo.tuwien.ac.at/), respectively. The radiosonde data and synoptic data were provided by National Centers for Environmental Information (NCEI) in the datasets of IGRA (https://www.ncei.noaa.gov/access/metadata/landing-



page/bin/iso?id=gov.noaa.ncdc:C00975) and ISD (https://www.ncei.noaa.gov/products/land-based-station/integrated-
surface-database), respectively. The enhanced GPS IWV dataset described in this paper is available at:
https://doi.org/10.5281/zenodo.6973528 (Yuan et al., 2022).

*Author contributions.* PY: Conceptualization, Methodology, Formal analysis, Investigation, Writing original draft, and
Visualization. GB, CK, WH, and DA: GPS ZTD Product generation. XY and PY: IGRA and ISD Data processing. RVM,
MM, WJ, and JA: Investigation and Reviewing. HK: Investigation, Reviewing, Supervision, Project administration, and
Funding acquisition. All: Manuscript editing.

*Competing interests.* The authors declare that they have no competing interests.

*Acknowledgements.* We are grateful to NGL for providing the GPS ZTD products and many institutions for sharing the
continuous GPS observations. We also thank ECMWF, TUW, and NCEI for providing the ERA5, TUW, IGRA radiosonde,
and ISD synoptic data. This work was funded at KIT with the German Research Foundation (DFG, project number:
321886779). This work was funded at NGL with NASA grants 80NSSC19K1044 and 80NSSC22K0463.





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





**Appendix A: Ranges of meteorological variables in 2020 from ERA5**

We determined the thresholds for the range checks of GPS IWV and ZTD by analysing their global variations in 2020 obtained from ERA5 atmospheric reanalysis. To reduce computational workload, here we used ERA5's surface level product instead of its pressure level product which was used in the development of the enGPS IWV product. ERA5's surface level product provides temperature, pressure, IWV and many other meteorological variables at the Earth's surface with a very high spatiotemporal resolution (0.25°×0.25° 1-hourly). For example, Figure A1a–b shows the minima and maxima of the 1-hourly IWV estimates at the ERA5 grids. Results show that the ERA5 IWV ranges from 0 to 94.9 kg m$^{-2}$. Hence, we set the lower and upper thresholds for the IWV range check as 0 and 100 kg m$^{-2}$, respectively.

Although the ZTD is not directly provided by ERA5, it can be calculated by using other meteorological variables included in ERA5. As we can know from Eq. (3), ZTD is comprised of ZHD and ZWD, and thus ZTD can be estimated if the two components are known. We calculated the ZHD at the grid nodes by using pressure from ERA5 and Eq. (5). The ZWD can be calculated by converting Eq. (1) to the form as follows:

$$ZWD = IWV \cdot R_V \cdot [k_2' + k_3/T_m],\tag{A1}$$

where the IWV is already available in ERA5 and $T_m$ can be calculated by using the surface air temperature ($T_s$) from ERA5 and the following equation (Bevis et al., 1992):

$$T_m = 70.2 + 0.72 \cdot T_s.\tag{A2}$$

The minima and maxima of the 1-hourly $T_m$ are shown in Fig. A1c–d. Results show that the global $T_m$ at the Earth's surface are between 208.9 and 304.9 K.

With the ZHD and ZWD obtained, the global ZTD estimates at the Earth's surface in 2020 were also calculated. The minimum and maximum of ZTD at each grid node are shown in Fig. A1e–f. The range of the ZTD is 1096.5–2835.8 mm. We therefore set the ZTD range as 1000– 3000 mm, which is wide enough to preserve the natural variations of ZTD at all the GPS stations.





**Figure A1.** Minima (a,c,e) and maxima (b,d,f) of IWV, $T_m$, and ZTD in 2020 obtained from 1-hourly 0.25°×0.25° ERA5 surface level product.



**Appendix B: Geopotential altitude from ellipsoidal height**

The vertical coordinate of a GPS station estimated by GPS data processing is usually ellipsoidal height ($H_{el}$, in metre), which is the height above reference ellipsoid. By contrast, the geopotential altitude (i.e., geopotential height; $H_{gp}$) in unit of geopotential metre (gpm) is referenced to Mean Sea Level (MSL) and adjusted according to gravity variations. As $H_{gp}$ is commonly used in Meteorology (e.g., ERA5 and radiosonde), we converted $H_{el}$ into $H_{gp}$ for the GPS stations.

We first converted the reference of the GPS station's altitude from ellipsoid to MSL by using geoid model EGM2008 (Earth Gravitational Model 2008; Pavlis et al., 2012):

$$H_{or} = H_{el} - N, \tag{B1}$$

where the $H_{or}$ and $N$ are the orthometric and geoid heights in metre, respectively.

We then adjusted the altitude of the GPS station (with a latitude of $\varphi$) by considering gravity variations (World Meteorological Organization, 2018):

$$H_{gp} = \frac{\gamma_s(\varphi)}{9.80665} \cdot \frac{R(\varphi) \cdot H_{or}}{R(\varphi) + H_{or}}, \tag{B2}$$

$$\gamma_s(\varphi) = 9.780325 \frac{1 + 1.93185 \times 10^{-3} \cdot \sin^2(\varphi)}{(1 - 6.69435 \times 10^{-3} \cdot \sin^2(\varphi))^{0.5}}, \tag{B3}$$

$$R(\varphi) = \frac{6.378137 \times 10^6}{1.006803 - 6.706 \times 10^{-3} \cdot \sin(\varphi)^2}. \tag{B4}$$

**Appendix C: Estimation of meteorological variables at GPS station**

In the development of the enhanced GPS (enGPS) IWV product, the required GPS station pressure ($P_s$) and weighted mean temperature ($T_m$) were calculated from the temperature ($T$), Relative Humidity (RH), and geopotential height (H) from ERA5 pressure level product.

Therefore, we first selected the four horizontally adjacent ERA5 grid nodes ($i, i = 1,2,3,4$) from the GPS station's location, and then carried out vertical adjustments and horizontal interpolations. For each horizontally adjacent grid node, the vertical adjustment was conducted as follows. If the station altitude ($H_s$) is lower than the bottom ERA5 pressure level at the $i$th node, then the station pressure was estimated with the International Civil Aviation Organization (ICAO) barometric correction formula:

$$P_s = P_0 \cdot \left(1 - \frac{\gamma}{T_0}(H_s - H_0)\right)^{\frac{g_0}{\gamma \cdot R_d}}, \tag{C1}$$

where $H_0$, $P_0$, and $T_0$ are the altitude, pressure, and temperature of the bottom ERA5 level at the $i$th node. $\gamma = 0.0065 \text{ K m}^{-1}$, $g_0 = 9.80665 \text{ m s}^{-2}$, and $R_d = 287.033 \text{ J kg}^{-1} \text{ K}^{-1}$.

If $H_s$ is between the $j$th and $(j + 1)$th ERA5 levels, we first estimated the station pressure from both levels with Eq. (C1) separately with values of $P_s^j$ and $P_s^{j+1}$. We then obtained the final adjusted station pressure by weighting of the two values (Schueler 2001):





$P_s = \frac{w_j}{w_j+w_{j+1}} P_s^j + \frac{w_{j+1}}{w_j+w_{j+1}} P_s^{j+1},$ (C2)

where $w_j = (H_j - H_s)^{-2}$ and $w_{j+1} = (H_{j+1} - H_s)^{-2}$.

As shown in Eq. (4) in Section 2.3, $T_m$ is dependent on the $T$ and water vapor pressure ($e$) profiles from station to the top. The $e$ was estimated by using $T$ and RH from ERA5:

$e = e_{sat}(T) \cdot \text{RH},$ (C3)

where $e_{sat}$ is saturation vapor pressure determined by Tetens formula provided by ECMWF's IFS Documentation CY45R1 (2018):

$e_{sat}(T) = a_1 \cdot \exp\left(a_3\left(\frac{T-T_0}{T-a_4}\right)\right),$ (C4)

where $a_1 = 6.112$ hPa and $T_0 = 273.16$ K. If $T \geq T_0$ (over water), $a_3 = 17.502$ and $a_4 = 32.19$ K. If $T \leq T_{ice}$ (250.16 K, over ice), $a_3 = 22.587$ and $a_4 = -0.7$ K. If $T < T < T_0$ (mixed phase), the $e_{sat}$ was estimated as a mix of those estimated

over water and ice ($e_{sat(water)}$ and $e_{sat(ice)}$):

$e_{sat} = e_{sat(ice)} + (e_{sat(water)} - e_{sat(ice)})\left(\frac{T-T_{ice}}{T_0-T_{ice}}\right)^2 .$ (C5)

In the end, we calculated the $P_s$ and $T_m$ by using a horizontal Inverse Distance Weighted (IDW) interpolation of the estimates at the four grid nodes (Jade and Vijayan, 2008).

In-situ meteorological observations were used to calculate $P_s$, $T_m$, and IWV at the GPS station in section 4. For

example, the $P_s$ was estimated by using synoptic observations from the ISD dataset with Eq. (C1). The $T_m$ was estimated with radiosonde observations from the IGRA dataset with Eq. (4) and Eqs. (C3)–(C5). The IGRA-derived IWV was calculated as follows:

$\text{IWV} = \frac{\sum_{H_s}^{H_{top}} q_j \Delta H_j}{\sum_{H_s}^{H_{top}} \frac{q_j}{g_j} \Delta H_j},$ (C6)

where $q_j$ (in kg kg$^{-1}$) and $g_j$ (in kg m$^{-2}$) are the average specific humidity and local gravitational acceleration between the

735 $j$th and $(j+1)$th levels from the station altitude to the top of the radiosonde profile. $\Delta H_j$ is the altitude difference between the two adjacent levels. The specific humidity was calculated as below:

$q = \frac{0.622e}{P-0.378e}.$ (C7)

The local gravitational acceleration was calculated as a function of latitude ($\varphi$ in rad) and orthometric altitude ($H_{or}$ in m) given by (World Meteorological Organization, 2018):

$g(\varphi, H_{or}) = 9.8062(1 - 2.6442 \times 10^{-3} \cos 2\varphi + 5.8 \times 10^{-6} \cos^2 2\varphi) - 3.086 \times 10^{-6} H_{or}.$ (C8)

**Appendix D: Vertical adjustment of ZHD in NGL's opGPS IWV product**

NGL employs the gridded ZHD product provided by TUW at a single level and calculate the ZHD at GPS stations by using vertical adjustment as follows (Boehm et al., 2007; Kouba 2008): 1) find the horizontally adjacent four grid nodes of a





particular GPS station and extract their ZHD values at the Earth's surface from the TUW product ; 2) convert the ZHD into

pressure ($P_0$) according to Eq. (4) at each node; 3) adjust the pressure from the altitudes of surface ($H_0$) to the GPS station ($H_s$) by using the barometric formula as follows:

$$P_s = P_0(1 - 2.26 \times 10^{-5}(H_s - H_0))^{5.225};$$    (D1)

4) convert the pressure at the altitude of station ($P_s$) over each grid node into ZHD with Eq. (5) in section 2.3; 5) estimate the ZHD at the location of the GPS station with a horizontal interpolation.