# Peer review of "An enhanced integrated water vapour dataset from more than 10,000 global ground-based GPS stations in 2020"

_Earth System Science Data, 2022_

## Referee Comment (RC2)

As expressed in the title, the manuscript developed a dataset which contains high quality integrated water vapour dataset from 12,552 ground-based GPS stations in 2020. Such dataset provides better accuracy in IWV than the current operational GPS IWV dataset also provide by Nevada Geodetic Laboratory. The enhanced IWV have been validated by comparing with the ones given by the operational GPS IWV and with nearby Radisonde-derived IWV. The error budget of enhanced IWV has been briefly discussed and the quality of the ERA5, which provides the weighted mean temperature and pressure for the conversion of ZWD to IWV, has also been validated by comparing with the RS data. Such dense network of enhanced IWV product is import to study both the temporal and spatial variations of the water vapour and is useful for a validation with IWVs given by other techniques. The manuscript is well written with a good structure. I would suggest that the manuscript should be published after a minor revision.

Some minor comments:

Line 44: remove "precise". radiosonde-derived IWV suffers from errors caused by the sensor characteristics that vary in time and space. It is better here just saying Radiosondes can provide vertical distribution of water vapour.

Line 69: "valule" → "value"

Line 88: Are those constraints recommended by GipsyX? Are those values suitable for all stations since you are processing data globally? Can you comment on such issue.

Line 98: "three-dimensional station coordinates every 24 hours" → "daily three-dimensional station coordinates"

Figure 1: In capital, "cyan" → "blue"

Line 108: "distances" –> "differences"

Line 135: "IWV" → "ZTD"

Line 154: "," → "."

Line 196: remove "-"

Figure 5: Use kgm$^{-2}$ as the unit of IWV in the figure to be consistent to the rest parts of the manuscript.

Line 305: "weak" → "small"

Line 309: "weak" → "small"

Line 317: "have" → "has"

Line 318: Change to "1 mm ZHD bias can result in a IWV bias ranging from

0.12 to 0.17 kgm$^{-2}$"

---

## Author Comment (AC1)

**Responses to Referee #1**

**https://doi.org/10.5194/essd-2022-274-RC1**

This paper is to present a high-quality global Integrated Water Vapour (IWV) dataset from 12,552 ground-based Global Positioning System (GPS) stations in 2020. Although the IWV data is useful, while it has no new methods or IWV improvement. Furthermore, it did not show new performances or achievements in weather prediction. Therefore it is against publication in the current form.

**Reply:** Thanks a lot for your affirmation on the usefulness of the dataset developed by this work. Our replies are shown with blue Arial font type. The related texts copied from the revised manuscript are shown with green Times New Roman font type.

Regarding your comment *of "it did not show new performances or achievements in weather prediction"*, it is noteworthy that this manuscript is a **data description paper** submitted to **Earth System Science Data (ESSD)** for the Special Issue of "**Analysis of atmospheric water vapour observations and their uncertainties for climate applications**". The aims and scopes of ESSD and the associated Special Issue are copied as follows:

**Aims and scopes of ESSD**

*https://www.earth-system-science-data.net/about/aims_and_scope.html*

*Earth System Science Data (ESSD) is an international, interdisciplinary journal for the publication of articles on __original research data (sets)__, furthering the reuse of high-quality data of benefit to Earth system sciences. The editors encourage submissions on __original data or data collections__ which are of sufficient quality and have the potential to contribute to these aims.*

*About this journal*

*This journal aims to establish a new subject of publication: to __publish data__ according to the conventional fashion of publishing articles, applying the established principles of quality assessment through peer review to data sets.*

**Manuscript types of ESSD**

*https://www.earth-system-science-data.net/about/manuscript_types.html):*

***Data description papers***

*These may __describe original research data, databases__, or combined datasets derived from them.*

*ESSD understands original research data as data generated by observation of the Earth system. In most cases this excludes reprocessing, postprocessing, or reanalysis of model outputs (such as climate model outputs, environmental modelling outputs, output of classification algorithms, and so forth). For the most part, ESSD does not publish model-based forecasts or projections. Articles may describe observational data extended or verified with model output or other combinations of model output and observational data produced for specific purposes or specific regions where observations remain too sparse or otherwise too limited. In all cases users must discover a unique accessible verifiable data product.*

*Although examples of data outcomes may prove necessary to demonstrate data quality, extensive interpretations of data – i.e. detailed analysis as an author might report in a research article – remain outside the scope of this data journal. ESSD data descriptions should instead highlight and emphasize the quality, usability, and accessibility of the dataset, database, or other data product and should describe extensive carefully prepared metadata and file structures at the data repository.*

*Articles in this category should not focus on instrumentation, methodology, data extraction, or data treatment except when that information helps quantify uncertainties or otherwise facilitates validation of data presented.*

*Please carefully read the ESSD data policy and editorial if unsure what kind of data are within the scope of the journal.*

**Aim of the special issue**

*https://essd.copernicus.org/articles/special_issue10_1118.html*

*The absorption of energy by water vapour keeps the average temperature on Earth at a level that makes life possible. However, the average temperature increased in recent decades due to increased greenhouse gas emissions. This anthropogenic increase is amplified by the water vapour feedback. Water vapour is further an important component of the water cycle and indirectly of the energy cycle. The role of water vapour in these cycles in a changing climate needs to be fully understood and quantified. Thus, **high-quality observations of water vapour** and the characterisation and validation of associated uncertainties are of high importance for climate applications. **Topics covered around atmospheric water vapour include, among others, retrieval development, data records, uncertainty characterisation and validation, inter-comparisons, climate applications, and model validation.** This special issue originates from the GEWEX Water Vapor Assessment (G-VAP, http://gewex-vap.org) and is open for all submissions within scope, not just from G-VAP.*

According to the scopes and aims of ESSD and the Special Issue, this work aims to overcome the limitations of previous GPS IWV datasets in quantity and quality, rather than achievements in weather prediction.

Regarding your comment of *"Although the IWV data is useful, while it has no new methods or IWV improvement"*, we cannot agree with it, because the dataset did show achievements in quantity as well as in quality.

This work provided an enhanced global dataset with 12,552 GPS stations through the collaboration with Nevada Geodetic Laboratory (NGL), USA, whereas most existing GPS Integrated Water Vapour (IWV) datasets only include several hundred stations. Moreover, compared to NGL's operational GPS IWV product, the quality of the GPS IWV dataset is enhanced by employing accurate meteorological information from ERA5 for the GPS IWV retrieval with a significantly higher spatiotemporal resolution. A dedicated data screening algorithm is also implemented.

The status, challenges, and the aims of this work are stated in Section 1 Introduction and copied as follows:

However, the insufficient spatial densities and coverages of current available GPS IWV datasets severely limit their capabilities in sensing Earth system processes over various spatial scales. Although approximately ten thousand of continuous dual-frequency GPS stations are currently in operation around the world, most existing regional and global GPS IWV datasets only include several hundred continuous stations or campaign measurements spanning less than one year, such as IGS (Wang et al., 2007; Heise 2009), EUREF (Pacione et al., 2017), HyMeX (Bock et al., 2016), EUREC4A (Bock et al., 2021), and GURN (Fersch et al., 2022). Combining their solutions for an integrated analysis is possible but it could be impacted by systematic differences due to various software packages and strategies used in their GPS data processing as well as approaches adopted for the conversion of IWV. Nevada Geodetic Laboratory (NGL) has taken a big step forward by integrating global GPS observations at more than 10,000 stations and providing their tropospheric ZTD and IWV estimates as well as other products (Blewitt et al., 2018). Although NGL processes the GPS observations uniformly by employing many state-of-the-art models and strategies, the accuracy of its operational IWV product is limited by the inaccurate auxiliary meteorological information used in IWV retrieval. An enhancement of NGL's operational GPS IWV product will provide more accurate IWV estimates and thus it can be more valuable for many applications, such as analysis of extreme weather and diurnal variations of IWV, and assessment of other IWV retrieval techniques.

The evaluations by using *in-situ* reference radiosonde and synoptic measurements did show the superiority of the enhanced GPS IWV product developed in this work compared to NGL's operational product. Details of the evaluations can be found in Section 5 Evaluation of enGPS versus opGPS. A short summary in the Abstract is copied as follows:

The enhancement is reached by employing accurate meteorological information from the fifth generation of European ReAnalysis (ERA5) for the GPS IWV retrieval with a significantly higher spatiotemporal resolution. A dedicated data screening algorithm is also implemented. The GPS IWV dataset has a good agreement with in-situ radiosonde observations at 182 collocated stations worldwide. The IWV biases are within $\pm 3.0$ kg m$^{-2}$ with a Mean Absolute Bias (MAB) value of 0.69 kg m$^{-2}$. The standard deviations (SD) of IWV differences are no larger than 3.4 kg m$^{-2}$. In addition, the enhanced IWV product shows substantial improvements compared to NGL's operational version, and it is thus recommended for high-accuracy applications, such as research of extreme weather events and diurnal variations of IWV, and intercomparisons with other IWV retrieval techniques. Taking the radiosonde-derived IWV as reference, the MAB and SD of IWV differences are reduced by 19.5% and 6.2% on average, respectively. The number of unrealistic negative GPS IWV estimates are also substantially reduced by 92.4% owing to the accurate Zenith Hydrostatic Delay (ZHD) derived by ERA5.

The dataset developed by this paper has wide potentials for various applications. For example, in this manuscript, we carried out intercomparisons of the GPS IWV with six different radiosonde types, which can be useful for the evaluations on possible wet/dry biases in specific radiosonde types. Particularly, the IWV intercomparisons between GPS and the radiosonde types of RS41 and Meisei were rarely reported as far as we know, and hence these results are valuable.

In addition, a potential application of this dataset is suggested by Frank Fell from the community comment (https://doi.org/10.5194/essd-2022-274-CC1):

> *We are currently establishing a total column water vapour data record from Sentinel-3 MWR observations covering the global ice-free global ocean.* ***The dataset presented by Yuan et al. has a potential to serve as ground-truth and therefore is of significant interest to our work, especially since it contains many GPS coastal stations.***

Therefore, we believe that this work fits scope of ESSD, especially the aim of this Special Issue. The dataset has been enhanced in not only quantity but also in quality, and thus it is beneficial to the community. With the free sharing of the dataset, the "*achievements in weather prediction*" and many other potential applications can be addressed not only by us but also by other international researchers in the future. We hope you can kindly reconsider this submission as a data description paper.

---

## Author Comment (AC2)

**Responses to Referee #2**

**https://doi.org/10.5194/essd-2022-274-RC2**

As expressed in the title, the manuscript developed a dataset which contains high quality integrated water vapour dataset from 12,552 ground-based GPS stations in 2020. Such dataset provides better accuracy in IWV than the current operational GPS IWV dataset also provide by Nevada Geodetic Laboratory. The enhanced IWV have been validated by comparing with the ones given by the operational GPS IWV and with nearby Radiosonde-derived IWV. The error budget of enhanced IWV has been briefly discussed and the quality of the ERA5, which provides the weighted mean temperature and pressure for the conversion of ZWD to IWV, has also been validated by comparing with the RS data. Such dense network of enhanced IWV product is import to study both the temporal and spatial variations of the water vapour and is useful for a validation with IWVs given by other techniques. The manuscript is well written with a good structure. I would suggest that the manuscript should be published after a minor revision.

**Reply:** Thanks a lot for your comments. Our replies are shown with blue Arial font type. The related texts copied from the revised manuscript are shown with green Times New Roman font type.

**Some minor comments:**

**1.** Line 44: remove "precise". radiosonde-derived IWV suffers from errors caused by the sensor characteristics that vary in time and space. It is better here just saying Radiosondes can provide vertical distribution of water vapour.

**Reply:** Removed as suggested.

**2.** Line 69: "valule" → "value"

**Reply:** We modified it as follows:

An enhancement of NGL's operational GPS IWV product will provide more accurate IWV estimates and thus **it can be more valuable** for many applications

**3.** Line 88: Are those constraints recommended by GipsyX? Are those values suitable for all stations since you are processing data globally? Can you comment on such issue.

**Reply:** In NGL's current GPS data processing strategy, the constraints of ZWD and two horizontal gradient parameters are set as 3 and 0.3 mm $h^{-1/2}$, respectively. These constraint values are employed according to Bar-Sever et al. (1998). We acknowledge that the optimal constraint values can be diverse in different regions under different weather conditions. Hence, NGL is addressing this issue as presented by Young et al. in the AGU Fall Meeting in December 2022. It is reported that using a loosed ZWD constraint of 6 mm $h^{-1/2}$ can generally reduce the variability of global GPS stations' vertical positions. Moreover, the variability of vertical positions

can be further reduced if the constrains are optimized individually for each station. It is therefore recommended to use station-specific optimal constraints of tropospheric delays in future.

We add the following sentences in the last paragraph of Section 6 Summary and outlook:

The GPS data processing strategy can also be further improved. For example, station-specific optimal constrains on the tropospheric delays and horizontal gradients can be applied to accommodate various weather conditions in different regions. Hence there will be a second-order enhancement of the IWV product. To do this goes beyond the scope of this work, but we suggest it as a promising extension of our investigation.

**Reference:**

Bar-Sever, Y. E., Kroger, P. M., and Borjesson, J. A.: Estimating horizontal gradients of tropospheric path delay with a single GPS receiver, Journal of Geophysical Research: Solid Earth, 103, 5019–5035, https://doi.org/10.1029/97JB03534, 1998.

Young Z., Blewitt G., and Kreemer C.: Application of Variable Random Walk Process Noise to Improve GPS Tropospheric Path Delay Estimation and Positioning at Local and Global Scales. https://agu.confex.com/agu/fm22/meetingapp.cgi/Paper/1162048. AGU Fall Meeting 2022 (Oral presentation).

**4.** Line 98: "three-dimensional station coordinates every 24 hours" → "daily three-dimensional station coordinates"

**Reply:** Modified as suggested.

**5.** Figure 1: In capital, "cyan" → "blue"

**Reply:** Modified as suggested. Moreover, in response to the comment from Frank Fell (Community), we divided the GPS stations into two subsets (coastal and inland stations) with a threshold of 20 km on their distances to sea. The coastal and inland GPS stations are displayed as blue and red dots in Fig. 1, respectively, as can be seen in our response to that comment.

**6.** Line 108: "distances" –> "differences"

**Reply:** We are sorry for the ambiguous description in the original manuscript, and we modified it as follows:

We blacklisted 57 stations as **the changes of their positions over time** were larger than 10 m in vertical or 100 m in horizontal.

**7.** Line 135: "IWV" → "ZTD"

**Reply:** Modified as suggested.

**8.** Line 154: "," → "."

**Reply:** Modified as suggested.

**9.** Line 196: remove "-"

**Reply:** Removed as suggested.

**10.** Figure 5: Use kgm⁻² as the unit of IWV in the figure to be consistent to the rest parts of the manuscript.

**Reply:** Modified as suggested. As we added a figure in the revised manuscript, the original Figure 5 is changed to Figure 6 as below:

[Figure]

**Figure 6.** Minima (a), maxima (b), mean values (c), and coefficients of variations (d) of the 5-min enhanced GPS IWV time series in 2020 at 9,418 stations. (e) IWV time series of four exemplary stations.

**11.** Line 305: "weak" → "small"

**Reply:** Modified as suggested.

**12.** Line 309: "weak" → "small"

    **Reply:** Modified as suggested.

**13.** Line 317: "have" → "has"

    **Reply:** Modified as suggested.

**14.** Line 318: Change to "1 mm ZHD bias can result in a IWV bias ranging from 0.12 to 0.17 kgm$^{-2}$ "

    **Reply:** Modified as suggested.

---

## Author Comment (AC3)

**Responses to Referee #3**

**https://doi.org/10.5194/essd-2022-274-RC3**

Comments on "An enhanced integrated water vapour dataset from more than 10,000 global ground-based GPS stations in 2020" submitted by Yuan, P., Blewitt, G., Kreemer, C., Hammond, W.C., Argus, D., Yin, X., Van Malderen, R., Mayer, M., Jiang, W., Awange, J., and Kutterer, H to "Earth System Science Data"

The study of IWV is an important topic in GPS, geodesy and other areas such as climate. This work did a comprehensive analysis of the two sets of IWV data from more than 10,000 GPS stations. One is from the Nevada Geodetic Laboratory (NGL) and the other is generated by the authors who used the European ReAnalysis (ERA5) for the GPS IWV retrieval. It showed that the IWV dataset generated by the authors had a better quality than that from the NGL. The authors did extensive analysis and evaluation of the two datasets using IWV data from 182 radiosonde stations.

**The comments are given below:**

**1.** Line 44 "For instance, satellite measurements have good spatial coverage, but their spatiotemporal resolutions could be low" This is not completely correct. The remote sensing satellite water vapor data can have spatial resolution of 1 km or even dozens of meter. The temporal resolution can also be dozens of minutes e.g. geostationary satellites.

**Reply:** Thank you for the information and we deleted the incorrect sentence. Based on the comments from you and Frank Fell (Community, https://doi.org/10.5194/essd-2022-274-CC1), we added the following sentence to Section 6 Summary and outlook:

The enGPS IWV product has broad potential applications. For instance, it has been employed in this study to evaluate the IWV measurements from specific radiosonde types. It can serve as ground-truth data for the validation of satellite water vapor retrievals.

**2.** Line 46, IWV should have its full spelling in its first use.

**Reply:** The full spelling of IWV was given in Line 40, where it occurs for the first time:

The atmospheric water vapour can be quantified as Integrated Water Vapour (IWV)

**3.** Line 111, the first citation of the Figure 1 is at line 111. However the Figure 1 is placed ahead of line 111. It is suggested to move Figure 1 after line 111.

**Reply:** Moved as suggested.

**4.** In Figure 1 and in the whole paper, only 182 radiosonde stations were used. There are many more radiosonde stations around the world. Why are only 182 radiosonde stations used?

**Reply:** This is because we only obtained 182 GPS-radiosonde-synoptic (GPS-RS-SYN) station clusters worldwide after careful checks on the data quality, availability, and the spatial and temporal matching between the global GPS, RS, and SYN stations.

In this study, the radiosonde observations are obtained from the Integrated Global Radiosonde Archive (IGRA) Version 2 provided by NOAA National Centers for Environmental Information (NCEI). NCEI collects global radiosonde and pilot balloon observations at more than 2,800 stations date back to the early 20th century (https://www.ncei.noaa.gov/access/metadata/landing-page/bin/iso?id=gov.noaa.ncdc:C00975). In the year of 2020, there were 770 stations in operation worldwide. As the radiosonde data are intended to calculate high-quality $T_m$ and IWV measurements serving as references for the evaluation of opGPS and enGPS IWV products, we employed the following quality control scheme as described in Section 2.6 In-situ meteorological observations:

IGRA is one of the most comprehensive RS datasets consisting of over 2800 stations worldwide, of which 770 are operational with regular daily observations at 00 and 12 UTC in the year of 2020. We used temperature and relative humidity profiles from the IGRA dataset and carried out quality control with the following criteria: 1) the profiles reach the surface and a pressure level of at least 300 hPa on top (Wang and Zhang, 2008); 2) data are available at no less than five (four) standard pressure levels for stations below (above) the 1000-hPa pressure level (Wang et al., 2005); 3) pressure gaps should be less than 200 hPa.

Likewise, we also provided more details on the pressure data from the ISD dataset:

ISD has more than 20,000 SYN stations worldwide and provides high-quality observations (e.g., pressure) with internal quality control procedures. In the year of 2020, the pressure observations are available at 11,637 ISD stations worldwide.

In order to evaluate the opGPS and enGPS IWV products and associated $T_m$ and ZHD, we employed high quality temperature and humidity profiles from the IGRA RS dataset as well as SYN pressure observations from the ISD dataset as references, respectively. To be comparable, the global GPS, RS, and SYN stations were matched in space and time as described in Section 4 Evaluation with in-situ observations:

As described in Sect. 2.6, there are 770 RS and 11,637 ISD stations available in 2020. In order to evaluate the opGPS and enGPS IWV products, we carried out a spatial and temporal matching of the GPS, RS, and ISD stations with the following criteria: 1) the distances of the RS and SYN stations with respect to their paired GPS stations are within 50 km in horizontal and 100 m in vertical (Wang and Zhang, 2008); 2) The time-matched GPS, RS and SYN observations are available at more than 50% of the 00 and 12 UTC epochs in 2020 and all the gaps in time are shorter than 30 days. The second criterion on data availability is necessary because the IWV differences from opGPS or enGPS with respect to RS can be seasonal dependent as discussed later in Section 5 (Fig. 13). The seasonal patterns can also be seen in the

differences in $T_m$ and ZHD. Therefore, the second criterion was applied to avoid incorrect explanations. It is noteworthy that 328 out of the 770 radiosonde stations had been discarded before the matching because their observations are only available at less than 50% of the 00 and 12 UTC epochs in 2020 or their gaps in time are longer than 30 days. With those criteria, we obtained 182 GPS-RS-SYN station clusters worldwide. The locations of the station clusters are listed in Table S1.

**Reference:**

Wang, J. and Zhang, L.: Systematic Errors in Global Radiosonde Precipitable Water Data from Comparisons with Ground-Based GPS Measurements, J. Climate, 21, 2218–2238, https://doi.org/10.1175/2007JCLI1944.1, 2008.

Wang, J., Zhang, L., and Dai, A.: Global estimates of water-vapor-weighted mean temperature of the atmosphere for GPS applications, J. Geophys. Res.: Atmos., 110, https://doi.org/10.1029/2005JD006215, 2005.

**5.** Line 650, "Yuan, P., Blewitt, G., Kreemer, C., Hammond, W.C., Argus, D., Yin, X., Van Malderen, R., Mayer, M., Jiang, W., Awange, J., and Kutterer, H.: An enhanced integrated water vapour dataset from more than 10,000 global ground-based GPS stations in 2020, https://doi.org/10.5281/zenodo.6973528, 2022." Why do you cite this? Your paper is citing your paper?

**Reply:** It is also unusual for us to cite the dataset in the list of references. However, this is the requirement from the submission guideline of Earth System Science Data:

*https://www.earth-system-science-data.net/submission.html*

*make sure to include the DOI (and/or review link) and citation of your dataset in the dedicated **data availability section** – e.g. https://doi.org/10.15770/EUM_SEC_CLM_1001 (Loew et al., 2015) – and add the corresponding entry to the **list of references** – e.g. Loew, A., Bennartz, R., Fell, F., Lattanzio, A., Doutriaux-Boucher, M., and Schulz, J.: Surface Albedo Validation Sites, EUMETSAT [data set], https://doi.org/10.15770/EUM_SEC_CLM_1001, 2015.*

**6.** The number of citations to your own papers is relatively high. Just cite closely relevant papers only.

**Reply:** We deleted six publications from us and kept only four closely relevant papers. In addition, the dataset developed by was included in the list according to the requirement from the journal as explained to your last comment.

Awange, J.: Environmental monitoring using GNSS: Global navigation satellite systems, Springer, 2012.

Awange, J.: GNSS environmental sensing, Springer, 2018.

Awange, J.: Food Insecurity & Hydroclimate in Greater Horn of Africa: Potential for Agriculture Amidst Extremes, Springer Nature, 2022.

Blewitt, G., Hammond, W. C., and Kreemer, C.: Harnessing the GPS data explosion for interdisciplinary science, Eos, 99, 485, 2018.

Jiang, W., Yuan, P., Chen, H., Cai, J., Li, Z., Chao, N., and Sneeuw, N.: Annual variations of monsoon and drought detected by GPS: A case study in Yunnan, China, Sci. Rep., 7, 5874, https://doi.org/10.1038/s41598-017-06095-1, 2017.

Van Malderen, R., Brenot, H., Pottiaux, E., Beirle, S., Hermans, C., Mazière, M. D., Wagner, T., Backer, H. D., and Bruyninx, C.: A multi-site intercomparison of integrated water vapour observations for climate change analysis, Atmos. Meas. Tech., 7, 2487–2512, https://doi.org/10.5194/amt-7-2487-2014, 2014.

Wagner, A., Fersch, B., Yuan, P., Rummler, T., and Kunstmann, H.: Assimilation of GNSS and Synoptic Data in a Convection Permitting Limited Area Model: Improvement of Simulated Tropospheric Water Vapor Content, Front. Earth Sci., 10, 2022

Yuan, P., Hunegnaw, A., Alshawaf, F., Awange, J., Klos, A., Teferle, F. N., and Kutterer, H.: Feasibility of ERA5 integrated water vapor trends for climate change analysis in continental Europe: An evaluation with GPS (1994–2019) by considering statistical significance, Remote Sens. Environ., 260, 112416, https://doi.org/10.1016/j.rse.2021.112416, 2021a.

Yuan, P; Kutterer, H: Point-scale IWV and zenith total delay (ZTD) derived for 66 stations of the global navigation satellite system (GNSS) Upper Rhine Graben network (GURN). https://doi.pangaea.de/10.1594/PANGAEA.936134, 2021.

Yuan, P., Van Malderen, R., Yin, X., Vogelmann, H., Awange, J., Heck, B., and Kutterer, H.: Characterizations of Europe's integrated water vapor and assessments of atmospheric reanalyses using more than two decades of ground-based GPS, Atmos. Chem. Phys. Discuss. [preprint], 1–38, https://doi.org/10.5194/acp-2021-797, in review, 2021b.

Yuan, P., Blewitt, G., Kreemer, C., Hammond, W.C., Argus, D., Yin, X., Van Malderen, R., Mayer, M., Jiang, W., Awange, J., and Kutterer, H.: An enhanced integrated water vapour dataset from more than 10,000 global ground-based GPS stations in 2020, https://doi.org/10.5281/zenodo.6973528, 2022.

**7.** In eq. (6), it seems you just consider the error in the conversion factor $\Pi$. Why didn't you consider the error in ZWD and its impact on IWV?

**Reply:** As shown in Eq. (3), the errors in ZWD are caused by the error in ZTD estimates from GPS data processing as well as the error in ZHD. Accurate pressure ($P_s$) measurement at the location of GPS stations is crucial for the calculation of its ZHD when using the Saastamoinen model (Eq. (5)). In addition to ZWD, $T_m$ is also important to the GPS IWV as can be seen from Eq. (1)−(2). In sum, the error sources of GPS IWV are the errors in ZTD from GPS data processing as well as the errors in ZHD ($P_s$) and $T_m$ from auxiliary meteorological data.

$$\text{IWV} = \Pi \cdot \text{ZWD}, \tag{1}$$

$$\Pi = \frac{10^6}{R_V \cdot [k_2' + k_3/T_m]}, \tag{2}$$

$$\text{ZWD} = \text{ZTD} - \text{ZHD}, \tag{3}$$

$$\text{ZHD} = 2.2768 \frac{P_s}{1 - 2.66 \times 10^{-3} \cdot \cos(2\varphi_s) - 2.8 \times 10^{-7} H_s}. \tag{5}$$

In this study, we discussed the impacts of errors in ZHD ($P_s$) and $T_m$ on the GPS IWV, because they are calculated in a more accurate way in the enhanced GPS (enGPS) IWV dataset compared to the ZHD and $T_m$ estimates used in NGL's operational GPS (opGPS) IWV dataset.

To be specific, the ZHD and $T_m$ in enGPS are calculated by using the latest ERA5 (0.25°×0.25°, 37 pressure levels, 1-hourly) with the consideration of vertical adjustment, whereas the ZHD and $T_m$ used in NGL's opGPS IWV product are obtained from Vienna University of Technology (TUW, https://vmf.geo.tuwien.ac.at/) with a much coarser resolution (2°×2.5°, surface level, 6-hourly).

It is noteworthy that both the enGPS and opGPS IWV datasets are based on the same GPS ZTD product provided by NGL, and hence we did not quantitatively investigate the errors in GPS ZTD when evaluating the superiority of enGPS IWV over opGPS IWV in Section 5 Evaluation of enGPS versus opGPS. More directly, we evaluated and compared the enGPS and opGPS IWV datasets by taking the independent IWV measurements from radiosonde (RS) as reference.

We noticed that there are several stations (e.g., station ASPA in Fig. 12j) with significant disagreements between their enGPS IWV and the reference RS IWV, but they are hard to be sufficiently explained with the errors in the associated ZHD ($P_s$) and $T_m$ evaluated by using with *in-situ* measurements from collocated synoptic stations and RS stations, respectively. Therefore, the discrepancies in their enGPS and RS IWV estimates are more likely due to station-specific errors in the GPS ZTD and RS IWV measurements. Accordingly, we modified the discussion on the IWV discrepancy at station ASPA in Section 5 Evaluation of enGPS versus opGPS:

Furthermore, the SD values of the RS-enGPS and RS-opGPS IWV differences are identical at station ASPA, with a value of 3.4 kg m$^{-2}$ (Fig. 12j). This is the maxima SD value over all the 182 station clusters in both enGPS and opGPS IWV datasets (Fig. 10b), indicating that the poor agreement could also be due to station-specific **errors in the GPS ZTD** and RS IWV measurements. Further comparisons with **other GPS ZTD solutions** and independent IWV measurements derived by other techniques will be helpful to explain the poor agreement, and it will be addressed in future studies.

**8.** Below 2.4 Screening of IWV, "The 5-min enGPS IWV data…" I am puzzled how you got the enGPS data. It is understandable you got opGPS data from NGL.

**Reply:** NGL produces GPS ZTD and IWV estimates operationally. However, its operational GPS (opGPS) IWV product is obtained based on the $T_m$ and ZHD products from TUM, which are with limited resolution (2°×2.5°, surface level, 6-hourly). The $T_m$ errors in NGL's opGPS IWV product could be large because no vertical adjustment to the altitude of GPS station was applied. Moreover, the ZHD errors in NGL's opGPS IWV product could be large due to its inaccurate vertical adjustment algorithm, especially in regions with complex terrains, see Appendix D: Vertical adjustment of ZHD in NGL's opGPS IWV product.

In this work, we tried to enhance the IWV (enGPS) dataset by using the auxiliary meteorological variables ($T_m$ and ZHD) from ERA5 pressure level product with improved resolution (0.25°×0.25°, 37 vertical levels, 1-hourly). We also used the state-of-the-art algorithms for the vertical adjustments of $T_m$ and ZHD (Appendix C: Estimation of meteorological variables at GPS station). Compared to NGL's opGPS IWV product, a global evaluation in Section 5.

Evaluation of enGPS versus opGPS shows that the enGPS IWV product developed in this work has better agreements with the *in-situ* $T_m$ and IWV measurements from the radiosonde stations, as well as the *in-situ* ZHD (pressure) from the synoptic stations.

The procedure of develop of the enGPS IWV product is as follows:

1. extract the GPS ZTD time series provided by NGL

2. remove the ZTD outliers with a range check and a moving window outlier check, see Section 2.2 Screening of Zenith Total Delay (ZTD)

3. calculate the meteorological variables ($T_m$ and ZHD) at the location of each GPS station from the 1-hourly ERA5 pressure level product (0.25°×0.25°, 37 vertical levels, 1-hourly), with the state-of-the-art vertical adjustment approaches, see Appendix C Estimation of meteorological variables at GPS station

4. interpolate the ERA5-derived $T_m$ and ZHD time series from 1-hourly to 5-min with a temporal linear interpolation at each station, and then use them for the conversion of enGPS IWV from NGL's ZTD, see Section 2.3 Retrieval approach of Integrated Water Vapour (IWV)

5. remove the IWV outliers with a range check and a moving window outlier check, see Section 2.4 Screening of IWV

In order to clarify the differences in the developments of NGL's opGPS IWV product and the enGPS IWV product developed in this work, we added a section (Sect. 2.5) and a flowchart (Fig. 4) in the revised manuscript as follows:

**2.5 Enhancements compared to NGL's operational IWV product**

Figure 4 compares the development processes of NGL's operational GPS (opGPS) IWV product and the enhanced GPS (enGPS) IWV product developed in this work. The enhancements are mainly in the auxiliary meteorological data for GPS IWV retrieval and the data screening.

[Figure]

**Figure 4.** Comparison of NGL's operational GPS (opGPS) IWV product and the enhanced GPS (enGPS) IWV product developed in this work.

The ZHD and $T_m$ used in NGL's operational IWV product were obtained from Vienna University of Technology (TUW, https://vmf.geo.tuwien.ac.at/) during the GPS data processing. TUW calculates the ZHD and $T_m$ every 6 hours at the Earth's surface with a 2°×2.5° grid based on an operational ECMWF product (Boehm et al., 2006). Although vertical adjustments of both ZHD and $T_m$ from the ECMWF surface level to the altitude of GPS station are essential for the accurate calculation of GPS IWV, only ZHD was vertically adjusted in NGL's opGPS IWV product with an empirical formular as described in Appendix D. The vertical change of $T_m$ has not been considered in the opGPS IWV product because an accurate lapse rate of $T_m$ is hard to obtain from the surface level product. In comparison, the enGPS IWV product was developed by using the meteorological data from the state-of-the-art ERA5 reanalysis with 1-hourly estimates at 0.25°×0.25° horizontal grids and 37 vertical pressure levels, with more advanced vertical adjustment correction algorithms as described in Section 2.3 and Appendix C. Moreover, despite non-real time, the atmospheric reanalysis products are generally considered to be superior to the operational analysis products due to improvements in input data and assimilation algorithms (Dee et al., 2009). Therefore, the meteorological data obtained from ERA5 are expected to be more accurate than those from TUW.

As for the quality control of the product, we carried out dedicated data screening (range check and outlier check) on the GPS ZTD and enGPS IWV time series as described in Section 2.2 and 2.4, respectively. However, these data screening steps were not included in the opGPS IWV product provided by NGL.

For a fair comparison to the enGPS IWV product in this work, we also screened the opGPS IWV product with the same algorithm in Section 2.4. The range check excludes 0.35% (3,849,630) of the datapoints with unrealistic negative IWV estimates, which is 13 times larger than that of enGPS. In particular, the IWV at four stations (CON2, MCG2, NAU2, and PHIG) on the high mountains (>3000 gpm) of Ross Island, Antarctica, were entirely smaller than zero in opGPS but rarely negative in enGPS. As an example, Figure 5 compares their IWV estimates at station PHIG (77.53°S, 167.05°E, 3185.9 gpm). All the opGPS IWV estimates are unrealistically below zero, with values ranging from -13.0 to -1.7 kg m-2. In contrast, the enGPS IWV are between -0.3 and 2.5 kg m-2 with negative values at only 0.32% (335 out of 105,389) of the datapoints. The negative opGPS IWV estimates at station PHIG are likely caused by the inaccurate ZHD provided by TUW with low spatial resolution and an unrealistic approximation of the barometric formula (Eq. (D1) in Appendix D) used to adjust pressure from the TUW model surface to an extremely high altitude. The results indicate that the enGPS IWV product is much superior to the

opGPS IWV product in avoiding unrealistic negative IWV estimates, especially at the stations with high altitudes. Therefore, the enGPS IWV product is recommended for those regions.

[Figure]

**Figure 5.** Comparison of the unscreened enGPS (blue) and opGPS (red) IWV at station PHIG (77.53°S, 167.05°E, 3185.9 gpm). $H_{gp}$ is geopotential altitude in geopotential metre (gpm; see Appendix B).

**9.** Line 462, it reads "Both the mean LMS6-enGPS IWV differences at daytime and nighttime are negative with values of -1.5 and -0.7 kg m-2, respectively." On line 464, it reads "By contrast, both the mean LMS6-enGPS IWV differences at daytime and nighttime are positive with values of 0.8 and 1.1 kg m-2, respectively." It seems these two are contradicting with each other.

**Reply:** Sorry for the typo. The latter one should be M10-enGPS. This part has been modified as follows:

Both the mean **LMS6-enGPS** IWV differences at daytime and nighttime are negative with values of -1.5 and -0.7 kg m$^{-2}$, respectively. Similarly, the values for Graw-enGPS IWV difference are -1.1 and -0.9 kg m$^{-2}$, respectively. By contrast, both the mean **M10-enGPS** IWV differences at daytime and nighttime are positive with values of 0.8 and 1.1 kg m$^{-2}$, respectively.

**10.** Line 489, "Finally, it is noted that the enhanced conversion of GPS-estimated ZTD to IWV does not affect GPS position estimates." It is not clear. Do you mean that the GPS positioning accuracy cannot be improved if the IWV is used in tropospheric error correction during GPS positioning calculation?

**Reply:** GPS positioning accuracy can be improved if the tropospheric delays are corrected with more accurate algorithms during GPS data processing. Here, we meant to remind the users of NGL's GPS position product that this study only enhanced the IWV product from NGL and we did not make any change on its GPS position product. As an outlook, we expect that both the accuracies of GPS position and ZTD (and the IWV consequently) estimates can be improved if tropospheric delays are corrected more accurately at the GPS data processing stage.

To be clearer, this sentence has been modified as follows:

Finally, it is noted that the implementation of the ERA5 meteorological variables in this work to enhance NGL's GPS IWV product was carried out after the GPS data processing, and hence it does not affect the operational GPS position product currently provided by NGL.

**11.** Line 490, "If higher resolution numerical models were implemented at the GPS data processing stage, then that should result in better position estimates together with the simultaneously estimated ZTD." What higher resolution numerical models are to be implemented? Higher temporal resolution or higher spatial resolution or both? What numerical models are you talking about? Do you talk the ERA5 model or other models?

**Reply:** We meant to talk about ERA5 and other models based on Weather Research and Forecasting Model (WRF). The newly released VMF3 developed by TU Wien (Landskron, and Boehm, 2018) is based on ECMWF operational Numerical Weather Model (NWM), and it includes mapping functions and zenith delays with a resolution of 6-hourly 1°×1° grids at surface level. In comparison, the ERA5 pressure level product has a resolution of 0.25°×0.25°, 37 pressure levels, and 1-hourly. Moreover, the regional NWMs for upper Rhine Graben developed by using WRF are with 1-hourly grids with spacing of 2.1 km and 72 vertical levels (Fersch et al., 2022). Owing to the extensive developments in NWM, the GPS tropospheric delays (ZTD, ZHD, and ZWD) are expected to be estimated more accurately at the GPS data processing stage, and thus the qualities of the GPS position and IWV estimates can also be enhanced.

This sentence has been modified as follows:

If higher resolution numerical models, such as ERA5 and the models based on Weather Research and Forecasting Model (WRF), were implemented at the GPS data processing stage, then that should result in better position estimates together with the simultaneously estimated ZTD.

**Reference:**

Landskron, D. and Boehm, J.: VMF3/GPT3: refined discrete and empirical troposphere mapping functions, J Geod, 92, 349–360, https://doi.org/10.1007/s00190-017-1066-2, 2018.

Fersch, B., Wagner, A., Kamm, B., Shehaj, E., Schenk, A., Yuan, P., Geiger, A., Moeller, G., Heck, B., Hinz, S., Kutterer, H., and Kunstmann, H.: Tropospheric water vapor: A comprehensive high resolution data collection for the transnational Upper Rhine Graben region, Earth Syst. Sci. Data. https://doi.org/10.5194/essd-14-5287-2022, 2022.

---

## Author Comment (AC4)

**Responses to Frank Fell (Community Comment)**

**https://doi.org/10.5194/essd-2022-274-CC1**

We are currently establishing a total column water vapour data record from Sentinel-3 MWR observations covering the global ice-free global ocean. The dataset presented by Yuan et al. has a potential to serve as ground-truth and therefore is of significant interest to our work, especially since it contains many GPS coastal stations.

**Reply:** Thanks a lot for your comments and we are glad to see that our dataset is beneficial to the community. We carried out a statistical analysis on the distance-to-sea (SeaDist) of the GPS stations and modified Fig. 1 as below, which could be interesting to you. The result shows that 3,745 out of the 12,555 (29.8%) GPS stations are located within 20 km of coastline.

[Figure]

**Figure 1.** (a) Geographical distribution of the 12,555 GPS stations and 182 radiosonde stations. (b–d) zoomed in figures for USA, Europe, and Japan, respectively. There are 3,745 GPS stations located within 20 km of coastline and they are labelled as Coastal GPS (blue dots), whereas the other 8,810 stations are labelled as Inland GPS (red dots). The different radiosonde types are labelled in various colours.